# Chromosome Xq23 is associated with lower atherogenic lipid concentrations and favorable cardiometabolic indices

Autosomal genetic analyses of blood lipids have yielded key insights for coronary heart disease (CHD). However, X chromosome genetic variation is understudied for blood lipids in large sample sizes. We now analyze genetic and blood lipid data in a high-coverage whole X chromosome sequencing study of 65,322 multi-ancestry participants and perform replication among 456,893 European participants. Common alleles on chromosome Xq23 are strongly associated with reduced total cholesterol, LDL cholesterol, and triglycerides (min $P = 8.5 \times 10^{-72}$), with similar effects for males and females. Chromosome Xq23 lipid-lowering alleles are associated with reduced odds for CHD among 42,545 cases and 591,247 controls ($P = 1.7 \times 10^{-4}$), and reduced odds for diabetes mellitus type 2 among 54,095 cases and 573,885 controls ($P = 1.4 \times 10^{-5}$). Although we observe an association with increased BMI, waist-to-hip ratio adjusted for BMI is reduced, bioimpedance analyses indicate increased gluteofemoral fat, and abdominal MRI analyses indicate reduced visceral adiposity. Co-localization analyses strongly correlate increased *CHRDL1* gene expression, particularly in adipose tissue, with reduced concentrations of blood lipids.

Mendelian, population, and functional genetic analyses of blood lipids (total cholesterol, low-density lipoprotein cholesterol [LDL-C], high-density lipoprotein cholesterol [HDL-C], and triglycerides) have yielded important fundamental insights regarding the root causes of coronary heart disease (CHD)[1,2]. For example, rare and common autosomal genomic variation influencing LDL-C, correspondingly influence CHD risk[3–6]. Such observations buttress clinical recommendations and bolster efforts to discover and validate lipid-related drug targets for CHD risk reduction[7–9].

Although the X chromosome comprises 5% of the genome, it has only been studied in a few genome-wide association analyses for blood lipids and coronary disease[10–13]. Major reasons for exclusion include incomplete coverage on genotyping arrays, potential discrepancies in genotyping quality on arrays due to haploinsufficiency in men, imputation and analytic challenges, and somatic X inactivation across tissues in women. Deep-coverage whole-genome sequencing (WGS) and analysis of the X chromosome now offers the promise for uniform coverage and high-fidelity genotyping for both sexes[14].

While differences in lipid levels and CHD risk by sex are well established[15,16], X chromosome dosage is also linked to lipid differences. Monosomy X (45X, Turner syndrome) is linked to dyslipidemia and premature CHD[17–19]. While obesity and gonadal deficiency was long believed to be the primary contributor to these phenotypes, women with Turner syndrome have higher total cholesterol, LDL-C, and triglyceride concentrations than age- and body composition-matched 46XX women with premature ovarian failure[19,20]. Men with an additional X chromosome (47XXY, Klinefelter syndrome) also suffer from infertility with higher rates of obesity, dyslipidemia, and CHD[21,22]. Furthermore, adult gonadectomized mice with XY, XX, and XXY chromosomes, regardless of gonadal sex, demonstrate dose-dependent changes in lipid levels[23]. Such observations, suggest that apparent sexual dimorphism in lipid levels may be explained by the sex chromosomes themselves.

Our study aims to discover X chromosome genomic variation associated with blood lipid levels among 65,322 multi-ancestry individuals with high-coverage whole X chromosome sequencing and available lipids in the NHLBI Trans-Omics for Precision Medicine (TOPMed) program[24]. Independent serial replication is performed in up to 390,606 and 66,287 individuals with GWAS array and lipids available in the UK Biobank and Nord-Trøndelag Health (HUNT) study, respectively[25,26]. We further evaluate the phenotypic consequences of lipid-associated variation in the UK Biobank, HUNT, and 176,899 additional participants of FinnGen[27]. Lastly, we perform colocalization analyses to pinpoint the possible causal gene in association regions. Here, we characterize an X chromosome locus associated with lipids and related cardiometabolic traits and prioritize CHRDL1 as the causal gene.

## Results

### Baseline characteristics, blood lipids, and chromosome X genotypes.
TOPMed sequences were aggregated and aligned, and variants were called by the TOPMed Informatics Research Center. A total of 65,367 out of 140,000 individuals in TOPMed freeze 8 with WGS data, including X chromosome sequence data had harmonized lipid levels available (Supplementary Fig. 1). Forty-five individuals with anomalous X chromosome copy number were excluded, leaving 65,322 individuals for analysis. 40,577 (62.1%) individuals were female and mean (standard deviation [SD]) age was 52.4 (14.9) years. Across all 21 included cohorts, 29,513 (45.2%) were white, 16,431 (25.2%) black, 13,432 (20.6%) Hispanic, 4714 (7.2%) Asian, 1182 (1.8%) Samoan, and 50 (0.1%)

Native American (Supplementary Table 1; Supplementary Fig. 2). The included studies were largely observational cohorts with some variations in ascertainment schemes as described in the Supplementary Note. Blood lipid distributions were generally similar across cohorts with some differences due to differences in study design and ancestry (Supplementary Table 2 and Supplementary Fig. 3). After adjusting for lipid-lowering medicines within each cohort and ancestry, we generated residuals within each cohort and race group adjusted for age, age[2], sex, 11 principal components of ancestry, and cohort-specific covariates. These residuals were inverse rank normalized and multiplied by the standard deviation within each cohort and race group to obtain effects in mg/dl units (see Methods) (Supplementary Fig. 4).

Among 65,322 TOPMed participants with lipid levels and WGS, we identified 19,898,222 total variants on the X chromosome by WGS. Of these variants, 88,008 (0.4%) were nonsynonymous variants and 4632 (0.02%) were rare (MAF < 1%) predicted protein-truncating variants. As expected, participants of African ancestry had the most X chromosome variants (Fig. 1a). Likely due to sample size differences, there were overall more total variants observed in our dataset among white participants compared to other ancestries (Fig. 1b). Within the X chromosome, females had a greater average [SD] number of variants per individual (133,255 [22,455]) than males (90,117 [12,166]), as expected (Supplementary Table 3). Generally, most of the variation observed across individuals was uncommon (i.e., 98.8% of variants had MAF < 5%) (Supplementary Table 4).

### X chromosome single-variant association with lipid levels.
In single-variant discovery analyses in TOPMed, we performed X chromosome-wide association analyses for genetic variants with minor allele count >20 that are not in the pseudoautosomal region, yielding 2.2 million analyzed of the 19.8 million detected. To maximize power, all samples (i.e., males and females) were included in the linear mixed model association analyses with SD-adjusted residuals of lipid levels as the outcome, where adjustments included sex (Supplementary Fig. 5).

Across variants assessed, we found 21 variants showing suggestive evidence ($P < 1 \times 10^{-6}$) of association with lipids in TOPMed (Supplementary Table 5 and Supplementary Fig. 6). We evaluated these associations for replication, serially, in the UK Biobank (N = 390,606) (Supplementary Table 6) and HUNT (66,635) (Supplementary Table 7). Three variants showed evidence of replication ($P < 0.05/21 = 0.002$) in UK Biobank and in HUNT and additionally met a stringent threshold for statistical significance in the meta-analysis (alpha = 0.05/2.2 M variants/4 traits = $5.7 \times 10^{-9}$) (Table 1).

The three variants occurred on chrXq23 and were all in at least moderate linkage disequilibrium across all included TOPMed participants (Supplementary Fig. 7 and Supplementary Table 8). They were also in moderate linkage disequilibrium with a previously described nearby variant, rs5985471[12], ($r^2$ 0.61–0.76). All three associated variants in our dataset have similar nonreference allele frequency (0.34–0.43), which was also similar between males and females. We observed similar associations for both males and females within TOPMed except male rs5985504-T carriers had greater decrease in triglycerides compared to female rs5985504-T carriers ($P_{interaction} = 0.001$) (Supplementary Table 9).

The minor alleles for these variants are common in all TOPMed ancestries except for Asian Americans (MAF 0.02) and Samoans (MAF 0.01). Nevertheless, effect estimates were largely of similar magnitude across ancestries in TOPMed for total

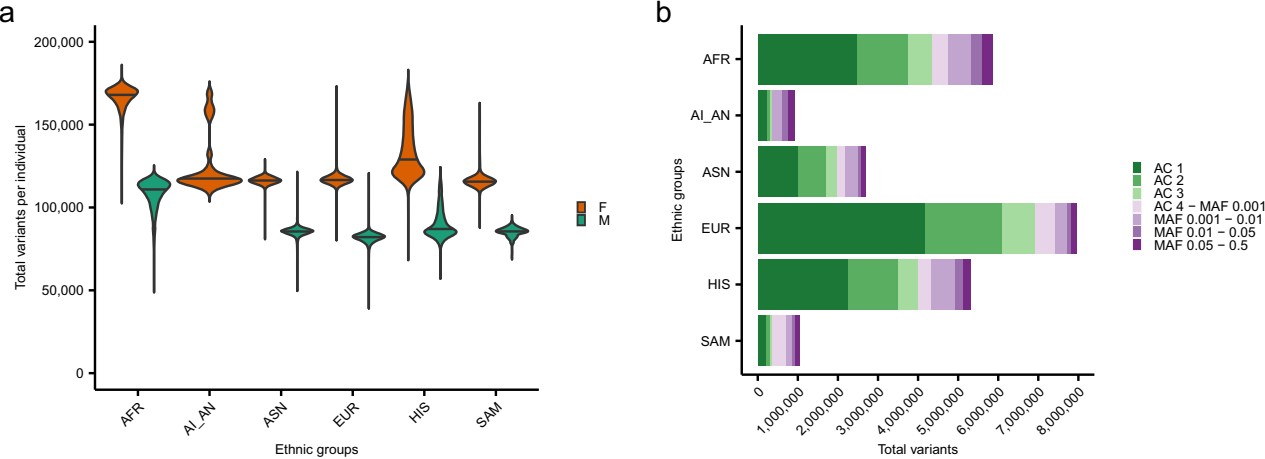

**Fig. 1 Distribution of X chromosome variants detected by whole-genome sequencing in TOPMed. a** Violin plots of the distributions of total X chromosome variants detected by whole-genome sequencing per sample by ancestry are depicted. Within each ancestry, distributions are shown by sex (orange: female; turquoise: male). Only discovery samples from TOPMed freeze 8 with lipids are included. **b** Across all TOPMed freeze eight samples with lipids, total X chromosome variants by ancestry are tabulated by allele count/frequency bins (dark green: AC 1; green: AC 2; light green: AC 3; lightest purple: AC 4—MAF 0.001; light purple: MAF 0.001–0.01; purple: MAF 0.01–0.05; dark purple: MAF 0.05–0.50). AC allele count, AI_AN American Indian / Native American / Alaskan Native, AFR African, ASN East Asian, EUR European, HIS Hispanic, MAF minor allele frequency, SAM Samoan, TOPMed Trans-Omics for Precision Medicine.

cholesterol (Supplementary Table 10) with no evidence of heterogeneity ($P_{heterogeneity} > 0.05$).

The three chrXq23 variants were associated with reduced atherogenic lipoproteins (i.e., total cholesterol, triglycerides, and LDL-C) (Table 1). The rs5942634-T allele is an intergenic variant and is 8 kb downstream from *RTL9* (also referred to as *RGAG1* in the literature), and was the top variant for total cholesterol, associated with 1.95 mg/dl lower concentration ($P = 2 \times 10^{-16}$). The rs5942648-A allele occurs 81 kb downstream, is intergenic between *RTL9* and *CHRDL1*, and was the top variant for LDL-C, associated with 1.53 mg/dl lower concentration ($P = 1 \times 10^{-12}$). The rs5985504-T allele resides 60 kb further downstream and is 68 kb from *CHRDL1* and was the top variant for log(triglycerides) leading to 2% lower triglycerides concentration ($P = 4 \times 10^{-11}$). Overall, the associated variants reside within a ~0.22 Mb linkage disequilibrium block spanning *RTL9* and *CHRDL1* (Supplementary Fig. 7). Within this block, variants within predicted active adult liver enhancers are in proximity to both the *RTL9* and *CHRDL1* genes (Supplementary Fig. 8). Only two variants reside within both an adult liver enhancer and DNase hypersensitivity site—rs2883091 in an intron of *RTL9*, and rs2143760 residing 4 kb from *CHRDL1* but 214 kb from *RTL9*. These variants are in at least moderate linkage disequilibrium ($r^2 > 0.60$) with the top associated variants in the locus. Virtual 4 C data additionally demonstrate a contact between the rs5985504 site and upstream of *CHRDL1* (Supplementary Fig. 9).

To determine whether our signal was independent of previously reported variants in the region, we performed conditional analysis for the associated between total cholesterol and rs5942634 with rs5943057[11], rs5985471[12], and rs5942937[13] (Supplementary Table 11). Previously reported SNPs were highly associated with total cholesterol when the variants were individually modeled. However, after adjusting for our reported total cholesterol variant (rs5942634), the known variants have dramatically lower effect estimates and are no longer associated with total cholesterol. On the other hand, rs5942634 remains marginally associated with total cholesterol and with a less of a change in effect size after adjusting for the three known variants. Similar results were obtained when adjusting the association between total cholesterol and rs5942634 for the individual

previously reported variants in the region (results not shown). This indicates that rs5942634 is only partially explained by the three reported variants.

**Phenome-wide association Of Chrxq23 variants.** Given prior genetic associations of LDL-C-lowering and triglyceride-lowering autosomal variants with lower risk for CHD, we hypothesized that sex chromosome variants lowering LDL-C or triglycerides would also lower risk for CHD. In HUNT, UK Biobank, and FinnGen (Supplementary Table 12), we observed that the top lipid-lowering alleles at this locus showed a reduced risk for CHD (Fig. 2). We found a 0.98 (95% CI 0.96, 0.99; $P = 1.7 \times 10^{-4}$) odds of CHD for each rs5942634-T allele, the lead cholesterol-lowering variant (alpha = 0.05 for the single haplotype assessment), and a correlation between the effect sizes of variants on total cholesterol in the chrXq23 locus and the effect sizes of these variants on CAD (r = 0.25), T2D (r = 0.33), and BMI (r = −0.34) (Supplementary Fig. 10).

To explore the range of phenotypes associated with the chrXq23 locus, we evaluated the associations of each of these three variants with 80 manually curated diverse clinical traits and conditions in the UK Biobank (Supplementary Table 13). Given the high degree of correlation among these variants, phenome-wide association results were similar (Supplementary Tables 14-16). As expected, the strongest associations were for reduced odds of hypercholesterolemia. Associations reaching a $P < 6.3 \times 10^{-4}$ ($P < 0.05/80$ traits) included reduced odds for diabetes mellitus type 2 (T2D), hypertension, and glaucoma, but increased odds for ever smoking as well as increased body-mass index (BMI) and body fat percentage. Notably, we observed lower odds of T2D for rs5942648 (OR = 0.97; 95% CI 0.96, 0.99; $P = 1.4 \times 10^{-5}$) (Fig. 2).

We additionally explored the association between each of these three variants with lipoprotein subspecies identified through nuclear magnetic resonance spectroscopy (NMR) within the Framingham Heart Study and Multi-Ethnic Study of Atherosclerosis cohorts (up to 6356 individuals). While we did not find any associations that passed a Bonferroni-corrected significance threshold ($0.05/(3 \text{ SNPs} \times 16 \text{ lipoprotein subspecies}) = 0.001$; Supplementary Table 17), we found two lipoprotein subspecies

**Table 1 Discovery and replication of chromosome Xq23 variants associated with lipid levels in TOPMed, UK Biobank, and HUNT.**

| rsID | Minor allele | Trait | Discovery TOPMed (N = 65,322) | | | | Replication UK Biobank Whites (N = 390,606) | | | UK Biobank non-Whites (N = 51,168) | | | HUNT (N = 66,287) | | | Meta-analysis | | | | |
|---|---|---|---|---|---|---|---|---|---|---|---|---|---|---|---|---|---|---|---|---|
| | | | MAF | Beta | SE | P | Beta | SE | P | Beta | SE | P | Beta | SE | P | Beta | SE | P | I² | P_meta |
| rs5942634 | T | TC | 34.4% | −1.95 | 0.24 | $2.0 \times 10^{-16}$ | −1.17 | 0.077 | $1.17 \times 10^{-52}$ | −1.043 | 0.23 | $6.47 \times 10^{-6}$ | −1.24 | 0.20 | $3.4 \times 10^{-10}$ | −1.23 | 0.066 | $3.78 \times 10^{-77}$ | 71% | 0.016 |
| | | log(TG) | | −0.017 | 0.0028 | $5.0 \times 10^{-9}$ | −0.025 | 0.0018 | $1.18 \times 10^{-44}$ | −0.027 | 0.0054 | $7.24 \times 10^{-7}$ | −0.011 | 0.0024 | $3.8 \times 10^{-6}$ | −0.020 | 0.0012 | $1.42 \times 10^{-56}$ | 88% | $1.90 \times 10^{-05}$ |
| | | HDL-C | | 0.14 | 0.084 | 0.09 | 0.12 | 0.026 | $8.99 \times 10^{-6}$ | 0.27 | 0.077 | $4.82 \times 10^{-4}$ | 0.068 | 0.062 | 0.27 | 0.13 | 0.022 | $8.63 \times 10^{-09}$ | 33% | 0.22 |
| | | LDL-C | | −1.53 | 0.22 | $2.0 \times 10^{-12}$ | −1.00 | 0.061 | $1.67 \times 10^{-60}$ | −1.028 | 0.18 | $2.39 \times 10^{-8}$ | −0.93 | 0.18 | $3.7 \times 10^{-7}$ | −1.028 | 0.053 | $1.31 \times 10^{-82}$ | 47% | 0.13 |
| rs5985504 | T | TC | 43.3% | −1.82 | 0.24 | $1.9 \times 10^{-14}$ | −1.18 | 0.077 | $5.37 \times 10^{-53}$ | −1.081 | 0.23 | $3.77 \times 10^{-6}$ | −1.30 | 0.20 | $3.3 \times 10^{-11}$ | −1.23 | 0.066 | $4.96 \times 10^{-78}$ | 57% | 0.072 |
| | | log(TG) | | −0.019 | 0.0029 | $4.2 \times 10^{-11}$ | −0.024 | 0.0018 | $2.36 \times 10^{-53}$ | −0.030 | 0.0055 | $3.39 \times 10^{-8}$ | −0.012 | 0.0024 | $4.3 \times 10^{-7}$ | −0.02 | 0.0013 | $1.29 \times 10^{-57}$ | 85% | 0.00022 |
| | | HDL-C | | 0.088 | 0.084 | 0.30 | 0.095 | 0.026 | $3.1 \times 10^{-4}$ | 0.25 | 0.078 | $1.13 \times 10^{-3}$ | 0.12 | 0.063 | 0.051 | 0.11 | 0.022 | $6.70 \times 10^{-07}$ | 18% | 0.30 |
| | | LDL-C | | −1.26 | 0.22 | $8.7 \times 10^{-9}$ | −0.99 | 0.061 | $3.87 \times 10^{-60}$ | −1.043 | 0.19 | $2.05 \times 10^{-8}$ | −1.00 | 0.18 | $3.0 \times 10^{-8}$ | −1.011 | 0.054 | $2.35 \times 10^{-79}$ | 0% | 0.70 |
| rs5942648 | A | TC | 38.3% | −1.88 | 0.24 | $8.2 \times 10^{-16}$ | −1.19 | 0.077 | $1.23 \times 10^{-53}$ | −1.13 | 0.23 | $8.21 \times 10^{-7}$ | −1.20 | 0.19 | $9.2 \times 10^{-10}$ | −1.24 | 0.066 | $1.69 \times 10^{-79}$ | 62% | 0.050 |
| | | log(TG) | | −0.016 | 0.0028 | $7.6 \times 10^{-9}$ | −0.025 | 0.0018 | $6.31 \times 10^{-45}$ | −0.030 | 0.0054 | $3.59 \times 10^{-8}$ | −0.012 | 0.0024 | $2.0 \times 10^{-6}$ | −0.02 | 0.0012 | $7.28 \times 10^{-58}$ | 88% | $2.17 \times 10^{-05}$ |
| | | HDL-C | | 0.14 | 0.083 | 0.094 | 0.11 | 0.026 | $2.77 \times 10^{-05}$ | 0.24 | 0.076 | $2.0 \times 10^{-3}$ | 0.090 | 0.063 | 0.15 | 0.12 | 0.022 | $4.71 \times 10^{-08}$ | 0% | 0.40 |
| | | LDL-C | | −1.53 | 0.22 | $1.2 \times 10^{-12}$ | −1.00 | 0.061 | $5.79 \times 10^{-61}$ | −1.076 | 0.18 | $3.47 \times 10^{-9}$ | −0.89 | 0.18 | $7.4 \times 10^{-7}$ | −1.028 | 0.053 | $1.02 \times 10^{-82}$ | 51% | 0.101 |

Variants attaining $P < 1 \times 10^{-6}$ in TOPMed (discovery) and $P < 0.002$ in UK Biobank and HUNT (replication studies). Effect estimates are presented from linear regression, adjusted for age, age², sex, batch, and principal components of ancestry, as well as cohort-specific covariates where appropriate. Two-sided $p$-values are presented; accounting for multiple-hypothesis testing, $p$-values $< 5.7 \times 10^{-9}$ are considered significant. rs5942634 was the top association for total cholesterol, rs5985504 for triglycerides, and rs5942648 for LDL-C. Since the variants were in at least moderate linkage disequilibrium (minimum pairwise $r^2 = 0.61$), there was evidence of association across all three of the aforementioned lipid traits for these three variants.
MAF minor allele frequency.

associated with suggestive evidence ($p < 0.05$), including greater concentration of medium HDL particles (but no effect on small or large HDL particles) and greater LDL size. We assessed for evidence of replication for indices related to LDL size (alpha 0.05) since the chrX variants associated with LDL-C. Among 6443 participants of the Atherosclerosis Risk in Communities cohort, we concordantly observed a $-0.034\,\mathrm{SD}$ ($P = 0.022$) lower concentration of small dense LDL for rs5942648-A. Among 365,365 participants of the UK Biobank, when using LDL-C/apolipoprotein B ratio as a proxy for LDL particle size, we observed a nominal increase in LDL size even with adjusting for both LDL-C and apolipoprotein B (Beta = $1.1 \times 10^{-5}$, $P = 0.048$).

To better characterize effects on adiposity given the aforementioned clinical phenotype associations, we evaluated the association between rs5942634-T and body composition measurements in the UK Biobank. Although rs5942634-T was associated with increased BMI, it was associated with slightly reduced waist-to-hip ratio adjusted for BMI (Beta = $-6.3 \times 10^{-4}$, SE = $1.1 \times 10^{-4}$, $P = 1.3 \times 10^{-8}$). rs5942634-T is associated with both increased truncal fat mass (Beta = 63 g, SE = 10 g, $P = 4.0 \times 10^{-10}$) as well as increased total peripheral fat mass, with increase of 21 g ($P = 3.6 \times 10^{-12}$) of the right leg, 20 g ($P = 3.4 \times 10^{-12}$) of the left leg, 7 g ($P = 4.1 \times 10^{-7}$) of the right arm, and 8 g ($P = 1.7 \times 10^{-9}$) of the left arm (Supplementary Table 18). Additionally, among 4750 unrelated UK Biobank participants with abdominal MRI measures available, rs5942634-T was associated with log-transformed inverse rank standardized increased abdominal subcutaneous adipose tissue (Beta = $+0.43$, SE = 0.15, $P = 5.9 \times 10^{-3}$) but decreased visceral adipose fat (Beta = $-1.12$, SE = 0.14, $P = 1.1 \times 10^{-15}$) to a greater degree. Given nine adiposity traits assessed, Bonferroni-corrected significance was assigned at $0.05/9 = 5.6 \times 10^{-3}$.

Rare pathogenic variants in *CHRDL1* were previously linked to X-linked recessive megalocornea, a condition characterized by enlarged corneal diameters with associated complications, including reduced visual acuity. Given these prior observations, we asked whether common variants associated with cholesterol at the *CHRDL1* locus were associated with differences in visual acuity. Among 112,842 UK Biobank participants (46.5% women; median age at assessment 58.5 years), we observed no association of lipid-associated chrXq23 alleles with altered visual acuity ($P > 0.05$; Supplementary Table 19). Given our sample size of 112,842 and SNP frequency of 34.4%, we had >99% power to detect effects >1/10th of a standard deviation unit of visual acuity at an alpha of 0.05.

**Gene expression analyses at chromosome Xq23**. We leveraged the GTEx eQTL data to better understand the gene or genes in the region that are influencing atherogenic lipid levels. Our most significant SNP, rs5942634, was associated with reduced expression of *CHRDL1* in skeletal muscle (beta = $-0.17$, $P = 1.2 \times 10^{-11}$), subcutaneous adipose (beta = $-0.16$, $P = 8.6 \times 10^{-8}$), visceral adipose (beta = $-0.17$, $P = 4.3 \times 10^{-6}$), and liver (beta = $-0.25$, $P = 5.9 \times 10^{-5}$). Additionally, rs5942634 was associated with increased expression of *RTL9* in skeletal muscle (beta = 0.18, $P = 2.7 \times 10^{-5}$; Supplementary Table 20).

Interrogating eQTL data for a single variant may lead to biased interpretations for causal gene inference. Therefore, we colocalized eQTL results for 8 genes (i.e., *ACSL4*, *TMEM164*, *AMMECR1*, *RTL9*, *CHRDL1*, *PAK3*, *CAPN6*, *DCX*) within the ChrXq23 region across prespecified lipid-related tissues (i.e., subcutaneous adipose, terminal ileum, visceral omentum adipose, whole blood, liver, and skeletal muscle) to relate aggregate blood lipid-association data with gene expression data. We observe that increased gene expression of *CHRDL1* shows consistent colocalization with decreased cholesterol

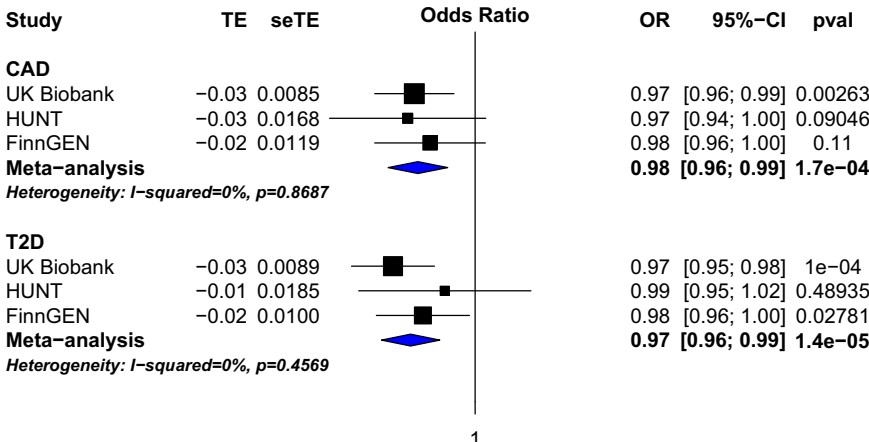

**Fig. 2 Association of lead cholesterol-lowering chrXq23 variant rs5942634-T with reduced odds of coronary heart disease and diabetes mellitus type 2.** The lead cholesterol-lowering allele at chrXq23 (i.e., rs5942634-T) and evidence of association with coronary heart disease and diabetes mellitus type 2 in each of three datasets in black, UK Biobank, HUNT, and FinnGEN, as well as meta-analysis in blue are shown. Odds ratios (OR) and 95% confidence intervals around the odds ratios are displayed.

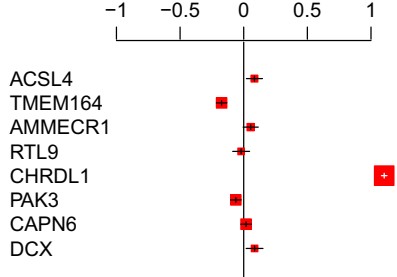

**Fig. 3 Colocalization of expression of genes at chrXq23 in subcutaneous adipose tissue with blood cholesterol effects strongly implicates *CHRDL1*.** The x-axis represents eight genes in the chrXq23 locus and y-axis represents standardized gene expression effect estimates in subcutaneous adipose tissues with 95% confidence intervals. Accounting for linkage disequilibrium, standardized effects and evidence of associations of cholesterol-lowering alleles were correlated with gene expression of genes at chrXq23 (*ACSL4, TMEM164, AMMECR1, RTL9, CHRDL1, PAK3, CAPN6, and DCX*).

across tissues, indicating that *CHRDL1* is the likely causal gene in the region (Fig. 3, Supplementary Fig. 11, and Supplementary Table 21).

**Rare chromosome X variant association analyses**. We performed SKAT gene-based tests of rare variants (MAF < 1%) across the X chromosome implemented by the GENESIS package within the TOPMed samples. We tested a maximum of 746 genes with more than 1 rare variant and at least 10 individuals carrying a minor allele with each lipid trait. No genes reached a Bonferroni-corrected significance threshold ($0.05/746 = 6.7 \times 10^{-5}$; Supplementary Table 22).

## Discussion

Using deep-coverage next-generation sequencing of the X chromosome in the NHLBI TOPMed program, we identified a locus that was previously associated with lipids in primarily European ancestry datasets, which we now extend to diverse ancestries along with strong replication in two independent studies. In addition to replicating the associations of chrXq23 lipid-lowering alleles with reduced odds for CHD, we also observe associations with reduced T2D odds and favorable adiposity indices. These

observations allow us to draw several conclusions about X chromosome genetic variation with blood lipid levels, as well as related cardiometabolic effects.

First, bioinformatic analyses implicate *CHRDL1* as a candidate causal gene for the association of chrXq23 variants with lipids. Based on genomic proximity of the strongest signal, *RTL9* was assigned as the likely causal gene in prior work[12]. However, colocalization analyses strongly prioritize increased *CHRDL1* gene expression in lipid-related, particularly adipose, tissues with reduced lipoprotein measures over other genes in the region. *CHRDL1* is not a previously known Mendelian lipid gene. In our study, disruptive rare coding variants in *CHRDL1* were not significantly associated with lipids, nor was any gene in the region.

Second, despite observing an association with increased BMI, chrXq23 lipid-associated alleles may lead to favorable effects on adiposity. We observed that chrXq23 lipid-lowering alleles were associated with increased gluteofemoral adipose tissue. Autosomal alleles similarly linked to expansion of gluteofemoral adipose tissue are associated with favorable risk for CHD and T2D[28,29]. Autosomal alleles associated with body fat distribution are also associated with various functional adipose measures, including morphology, lipolysis, and lipogenesis[30]. Recent gene expression analyses of human adipose-derived stromal cells showed persistent upregulation of *CHRDL1* after inducing adipogenesis[31]. *CHRDL1* is believed to influence adipogenic differentiation in human isolated preadipocytes[32]. Comparative gene expression analyses suggest relatively greater *CHRDL1* expression in subcutaneous versus visceral fat[32,33]. In our analyses, the chrXq23 lipid-lowering alleles were associated with an increase in abdominal subcutaneous adipose tissue but a decrease in visceral adipose tissue. A proteomic discovery analysis showed that increasing circulating *CHRDL1* concentrations were associated with increased birth weight but decreased triglycerides and homeostatic model assessment of insulin resistance[34].

Third, chrXq23 lipid-lowering alleles have favorable cardiometabolic effects that appear to reduce risk for CHD and T2D. Our results for CHD at chrXq23 are consistent with prior work at this locus[12], and extend to prior observational epidemiology, genetic, functional, and clinical trial evidence implying a causal relationship between reduced LDL-C and reduced CHD risk[35]. In aggregate, autosomal LDL-C-reducing alleles are associated with increased T2D risk but the effects are inconsistent across individual variants[7,36,37]. In meta-analyses of randomized controlled trials, statins are associated with a modestly increased risk of

incident T2D[38,39]. The effects of triglyceride-lowering alleles and T2D risk are generally inconsistent[40–42]. Common triglyceride-lowering variants at *LPL* and *ANGPTL4* p.E40K in the lipoprotein lipase pathway are associated with reduced triglyceride concentrations, CHD odds, and T2D risk[40,43,44]. Rare loss-of-function variants in *ANGPTL3*, also implicated in the lipoprotein lipase pathway, are associated with reduced LDL-C and triglyceride concentrations as well as reduced CHD odds but favorable effects on T2D risk have not yet been observed[40,45,46]. However, we uniquely describe a genetic locus associated with reduced LDL-C concentrations, reduced triglyceride concentrations, lower CHD risk, and lower T2D risk. Similarly, we observed that lipid-lowering chrXq23 alleles were independently associated with increased LDL-C/apoB ratio, which has been independently associated with reduced CHD and T2D risk in observational epidemiologic studies[47–50]. These data imply that therapeutic modulation of the causal pathway may lead diverse favorable cardiometabolic indices. Whether implicated variants influence the lipoprotein lipase pathway or represent a novel lipid-related pathway for combined CHD and T2D should be addressed by future research.

Fourth, common lipid-associated variants linked to increased *CHRDL1* expression are not associated with visual acuity measures. Ventropin, the product of *CHRDL1*, was first described as a bone morphogenic protein 4 inhibitor and a regulator of retinal development[51]. Pathogenic disruptive variants in *CHRDL1* are implied in X-linked megalocornea[52]. However, common lipid-associated variants linked to increased *CHRDL1* expression in the present study do not associate with measures of visual acuity in the UK Biobank. These data imply that therapeutic modulation to recapitulate the protective effects associated with chrXq23 lipid-lowering alleles is not anticipated to lead to on-target adverse visual acuity effects.

Important limitations should be considered in the interpretation of our findings. First, our genetic association analyses of the X chromosome do not account for random X inactivation. Accounting for random X inactivation is expected to modestly improve power and thus our approach biases our findings toward the null[53,54]. We found that there was slightly higher variance of total cholesterol in heterozygous females (sd = 44.1 mg/dl) compared to homozygous females (homozygous ref sd = 43.0, homozygous alt sd = 43.9) of rs5942634 using the Levene's Test for Homogeneity of variance ($P = 0.002$), indicating this locus may be subject to random X inactivation. Second, while our in silico analyses and prior literature strongly implicate *CHRDL1* as the causal gene, additional analyses including perturbational experiments in model systems are necessary to confirm our hypotheses. Furthermore, whether additional *cis*-acting or *trans*-acting gene expression for other genes are additionally relevant for the observed effects on blood lipids are currently unknown. Third, chrXq23 lipid-lowering alleles were associated with both increased truncal and gluteofemoral adiposity indices, and whether the former associations result in adverse clinical consequences is not known. Nevertheless, phenome-wide association analyses did not reveal concerning clinical phenotype associations related to modest increases in BMI. Notably, chrXq23 lipid-lowering alleles were associated with decreased visceral adipose tissue.

In conclusion, we observe a consistent association of chrXq23 alleles with reduced total cholesterol, LDL-C, triglycerides, CHD, and T2D. Despite an increase in BMI, these alleles were favorably associated with increased gluteofemoral and abdominal subcutaneous adiposity, decreased visceral adiposity, and increased LDL-C/apolipoprotein B ratio. Colocalization analyses strongly implicate increased *CHRDL1* expression in adipose tissues with these favorable cardiometabolic indices, pointing to *CHRDL1* as the leading candidate gene in the region.

## Methods

**Study participants**. For discovery, 65,322 individuals from 21 studies in the freeze 8 release of the NHLBI TOPMed Program with WGS passing central quality control by the TOPMed Informatics Research Core and blood lipid data available were included for analysis (Supplementary Fig. 1). The included studies are Atherosclerosis Risk in Communities study (ARIC, 7991), Old Order Amish (Amish, 1083), Mt Sinai BioMe Biobank (BioMe, 9857), Coronary Artery Risk Development in Young Adults (CARDIA, 3054), Cleveland Family Study (CFS, 577), Cardiovascular Health Study (CHS, 2773), Diabetes Heart Study (DHS, 365), Framingham Heart Study (FHS, 3990), Genetic Epidemiology Network of Arteriopathy (GENOA, 1,044), Genetics of Lipid-Lowering Drugs and Diet Network (GOLDN, 924), Genetic Epidemiology Network of Salt Sensitivity (GenSalt, 1770), Genetic Studies of Atherosclerosis Risk (GeneSTAR, 1755), Hispanic Community Health Study—Study of Latinos (HCHS/SOL, 7391), Hypertension Genetic Epidemiology Network and Genetic Epidemiology Network of Arteriopathy (HyperGEN, 1853), Jackson Heart Study (JHS, 2846), Multi-Ethnic Study of Atherosclerosis (MESA, 5283), Massachusetts General Hospital Atrial Fibrillation Study (MGH_AF, 683), San Antonio Family Study (SAFS, 617), Samoan Adiposity Study (Samoan, 1182), Taiwan Study of Hypertension using Rare Variants (THRV, 1976), and Women's Health Initiative (WHI, 8305) (Please refer to the Supplementary Note for additional details). Study participants provided consent per each study's IRB approved protocol. These data were secondarily analyzed through a protocol approved by the Partners Healthcare IRB and Boston University IRB.

For lipid replication, 69,635 participants from the Nord-Trøndelag Health (HUNT) study and 390,606 participants of the UK Biobank with genome-wide array data and lipid data were included. The HUNT study is a longitudinal, repetitive population-based health survey conducted in the county of Nord-Trøndelag, Norway[26]. Since 1984, the adult population in the county has been examined three times, through HUNT1 (1984–86), HUNT2 (1995–97), and HUNT3 (2006–08). A fourth survey, HUNT4 (2017–2019), is ongoing. HUNT was approved by the Data Inspectorate and the Regional Ethics Committee for Medical Research in Norway (REK: 2014/144). All HUNT participants gave informed consent. Approximately 120,000 individuals have participated in HUNT1–HUNT3 with extensive phenotypic measurements and biological samples. The subset of these participants that have been genotyped (~70,000) using Illumina HumanCoreExome v1.0 and 1.1 and imputed with Minimac3 using a combined HRC and HUNT-specific WGS reference panel are included in the current study. The UK Biobank is a large, prospective cohort of ~500,000 United Kingdom residents aged 40–69 years[25,55]. Patients provided answers to questions regarding socio-demographic, lifestyle, and health-related factors; additionally, participants provided blood, urine, and saliva for genetic and other future assays. Genotyping was performed on a custom Affymetrix array followed by imputation. Various additional measurements were performed on all recruited participants (e.g., electrocardiography, etc) and some measurements in subsets (e.g., cardiac magnetic resonance imaging, etc). Study participants provided consent per the UK Biobank's IRB approved protocol. We excluded UK Biobank individuals that met the following criteria: (1) Individuals whose submitted gender is not same as inferred gender; (2) Individuals with putative sex chromosome aneuploidy; (3) Individuals with second degree or higher degree relatives; and (4) Individuals who withdrawn consent. We analyzed individuals who were British white separately from those that were not considered British White. These data were secondarily analyzed through a protocol approved by the Partners Healthcare IRB.

The effects of lipid-associated X chromosome variants on CHD risk were estimated in 69,635 participants of HUNT, 390,606 British White participants of UK Biobank, and 176,899 participants of FinnGen. The effects on risk of diabetes mellitus were estimated in 69,635 participants of HUNT, 390,606 participants of UK Biobank, and 171,087 participants of FinnGen. FinnGen (https://www.finngen.fi/en) is a large biobank study that aims to genotype 500,000 Finns on a FinnGen ThermoFisher Axiom custom array, with the current data freeze comprising 181,820 Finnish individuals[27]. FinnGen includes prospective epidemiological and disease-based cohorts, and hospital biobank samples. The data were linked by the unique national personal identification numbers to national hospital discharge (available from 1968), death (1969-), cancer (1953-), and medication reimbursement (1995-) registries and disease endpoints were defined by harmonizing the International Classification of Diseases (ICD) revisions 8, 9, and 10, cancer-specific ICD-O-3 and ATC-codes. The FinnGen project is approved by Finnish Institute for Health and Welfare (THL).

**Sequencing, genotyping, and quality control of TOPmed**. Whole-genome sequencing of at least 30× was performed across six sequencing centers using PCR-free library preparation kits for TOPMed samples[24]. In most cases, all samples for a given study were sequenced at the same center. Samples were excluded if estimated contamination by verifyBamId was >3%{Jun, 2012 #278} or <95% of the genome attained >10× coverage. The reads were centrally realigned to human genome build GRCh38 at each center using BWA-MEM[56,57]. Joint variant discovery was subsequently performed with the 'GotCloud' pipeline by the IRC[58]. The variant calling software tools are under active development; updated versions can be accessed at http://github.com/atks/vt, http://github.com/hyunminkang/apigenome, and https://github.com/statgen/topmed_variant_calling. Using sequencing quality metrics, a catalogue of

previously discovered variants, and variants with Mendelian inconsistencies from included families, variant-level quality controlled was performed using a support vector machine algorithm. Sample-level quality control was performed by the TOPMed IRC and Data Coordinating Center (DCC) to remove samples genotypic/reported inconsistencies for pedigree and sex, and substantial discordance with prior genome-wide array genotyping. Only variants and samples that passed quality control were included in the call set.

One individual from duplicate pairs identified by the DCC was removed, retaining the individual with lipid levels available when one did not have lipid levels. If both individuals had lipid levels, one individual was randomly selected. Individuals were excluded when their genotype determine sex did not match phenotype reported sex ($n = 6$) and individuals <18 years old where excluded ($n = 865$).

**Blood lipid measurements and phenotypic modeling.** Conventionally measured fasting blood lipids, including total cholesterol, LDL-C, HDL-C, and triglycerides, were included for analysis. Harmonization of the lipid values, lipid-lowering medication status, and fasting status at lipid blood draw was performed by the TOPMed Data Coordinating Center. LDL-C was either calculated by the Friedewald equation when triglycerides were <400 mg/dl or directly measured. Given the average effect of lipid-lowering medicines, when lipid-lowering medicines (largely statins) were present, total cholesterol was adjusted by dividing by 0.8 and LDL-C by dividing by 0.7, as previously done[2]. Triglycerides were natural log transformed for analysis. Standard deviation scaled inverse-normalized residuals adjusting for age, age[2], sex, the first 11 PCs of ancestry (as recommended by the TOPMed DCC), as well as cohort-specific covariates (study site or known founder mutations), where created within each study by self-reported race. Effect sizes are reported in mg/dl or log(mg/dl) for TG.

**Coronary heart disease and diabetes mellitus type 2 phenotyping.** In the UK Biobank, we used National Health System OPCS-4 (Office of Population, Census and Surveys: Classification of Interventions and Procedures, version 4) codes K40.1-40.4, K41.1-41.4, K45.1-45.5, K49.1-49.2, K49.8-49.9, K50.2, K75.1-75.4, or K75.8-75.9 to indicate the presence of coronary heart disease. For diabetes mellitus type 2 classification in the UK Biobank, we used the presence of OPCS-4 codes E11.0-E11.9 or ICD9 code 1223.

Coronary heart disease (CHD) in HUNT was defined as individuals with self-reported coronary artery bypass, angioplasty, or stent placement or with diagnosis of myocardial infarction or chronic ischemic heart disease based on at least one occurrence of the following codes: ICD9: 410, 411, 412, 414.0, 414.8, 414.9 or ICD10: I21, I22, I23, I24, I25.1, I25.2, I25.5, I25.6, I25.7, I25.8, I25.9. Individuals with angina were excluded from controls. Type II diabetes was defined based on at least 1 occurrence of the following diagnosis codes: ICD9: 250.00, 250.02, 250.10, 250.12, 250.20, 250.22, 250.30, 250.32, 250.40, 250.42, 250.50, 250.52, 250.60, 250.62, 250.70, 250.72, 250.80, 250.82, 250.90, 250.92 ICD10: E11 or by diagnosis of diabetes during HUNT clinical examinations (nonfasting serum or blood glucose > 11.1 mmol/L or Hemoglobin A1C > 6.5%).

In FinnGen, CHD cases were defined as subjects with either an underlying or direct cause of death with ICD codes I20-I25, I46, R96 or R98 (ICD10) or 410–414 or 798 (ICD9/8), a hospital discharge diagnosis with ICD codes I200, I21-I22 (ICD10) or 410, 4110 (ICD9/8) and/or a coronary revascularization procedure (coronary artery bypass surgery procedure or coronary angioplasty, or an entry of invasive cardiac procedures in the country-wide register. Type 2 diabetes was defined as subjects with an underlying or direct cause of death or as the main or side diagnosis at hospital discharge with ICD codes E11 (ICD10)/250(0–9)A (ICD9) with ICD codes E11 (ICD10); 250(0–9)A (ICD-8/9) or at least three prescription medicine purchases with ATC class A10B, or as the specially reimbursed medication for diabetes. Cases with both type 2 and type 1 diabetes mellitus were excluded. In these definitions, the ICD10, ICD9 and ICD-8 below refer to the Finnish versions of the ICD codes.

**UK Biobank phenotype ascertainment.** Our approach to phenome-wide association analyses are similar to prior efforts[59,60]. Briefly, a phenome-wide association analysis was performed to evaluate the associations of X chromosome lipid-associated variants with a broad range of clinical phenotypes[61]. A total of 80 manually curated traits were classified according to a combination of self-report and billing codes, except for the following which were based on corresponding UK Biobank data fields: death (40000), ever smoked (20610), BMI (23104), and percentage body fat (23099) (Supplementary Table 13). Lipid-associated variants were associated with each trait, using linear regression for continuous traits and logistic regression for dichotomous traits, adjusting for age, sex, array type, and the first five PCs in the model.

For adiposity analyses, body composition values were obtained using bioelectrical impedance measurement (Category 100009) with a Tanita BC418MA body composition analyzer. Separate readings for fat percentage, mass, and free mass as well as predicted muscle mass are generated for the whole body, trunk, each leg, and each arm. In linear regression analyses, lipid-associated variants were associated with fat mass (in kilograms) for each of the aforementioned components adjusting for age, sex, and the first five PCs in the model. We also separately

analyzed the association of lipid-associated variants with abdominal subcutaneous adipose fat (22408) and visceral adipose fat (22407) among unrelated UK Biobank participants with abdominal MRI measures available[62]. Each of these phenotypes was natural log-transformed and inverse rank standardized; linear regression models were then adjusted for age, sex, array type, and the first 11 PCs.

For visual acuity analyses, we included data where baseline visual acuity was available for at least one eye and genotyping data was available in the UKBB[63]. Methods from visual acuity assessment in UKBB were previously described. Briefly, visual acuity was measured using the logarithm of the minimum angle of resolution ("LogMAR") chart (Precision Vision, LaSalle, Illinois, USA) at a distance of four meters. Using one eye at a time, participants were tasked with identifying the five letters displayed at the top row; they proceeded to read letters from successive rows, which had progressively smaller text. The test was terminated when two or more of the five letters on a given row were read incorrectly. Visual acuity was computed based on the number of successfully read rows; the visual acuity result was provided by the UK Biobank as Field ID 5208 (left eye) and Field ID 5201 (right eye). Data from the right eye was available in 60,421 women and 51,245 men, while data from the left eye was available in 60,402 women and 51,280 men. Linear mixed models from the lme4 package in R (v3.6.1) were used to estimate the association if each of the three variants with visual acuity, adjusting for age, sex, the first five PCs in the model, as well as a random effect accounting for which eye was tested (left or right).

**Single-variant association analyses.** For discovery, each single variant on the X chromosome with at least 20 copies of the minor allele was analyzed for association with each adjusted blood lipid residual across all TOPMed samples with lipid levels available (see Blood lipids measurements and phenotypic modeling) using a fast linear mixed model with kinship adjustment (SAIGE-QT, version 0.29.4.4[64]) since a large proportion of TOPMed participants are related in ENCORE (https://encore.sph.umich.edu) additionally adjusting for the first 11 PCs in the model. SAIGE-QT was specifically used to maximize computational efficiency given hosting and kinship precomputation by the TOPMed Informatics Research Core. Heterozygous and homozygous women were coded as having 1 or 2 nonreference alleles, respectively. Hemizygous males were coded as having two reference alleles. This modeling is consistent with random X inactivation of one of two X chromosomes in females yet consistent expression of the single X chromosome in males.

For SNPs with suggestive evidence of association in TOPMed ($P < 1 \times 10^{-6}$), we sought replication in UKBB. For replication in UKBB unrelated individuals, we performed linear regression associations using R version 3.6.0. Covariates included age, age[2], sex, the first 10 PCs in the model, and genotyping array.

For replication in HUNT, a cohort within a founder population, plasma lipids were analyzed using efficient linear mixed models implemented by BOLT-LMM v2.3.1[65] with covariates for sex, age, age[2], batch, and principle components 1–4. CAD was analyzed using SAIGE with birth year, batch, sex, and principle components 1–4 as covariates.

Covariates included in the models of association for each contributing study were based on study characteristics and recommendations from study investigators. We took loci reaching a suggestive association with lipid levels ($P < 1 \times 10^{-6}$) in TOPMed onto replication in UKBB and HUNT. For replication, we used a significance level of 0.002 (Bonferroni correction for 21 loci) that met a suggestive level of association in TOPMed. We used a fixed effects meta-analysis to combine the association results from TOPMed, UKBB, and HUNT. We set the statistical significance for our meta-analysis to be alpha = $5.7 \times 10^{-9}$ (0.05/2.2 M variants/4 traits = $5.7 \times 10^{-9}$), which is more stringent than a standard genome-wide significance threshold of $5 \times 10^{-8}$. Heterogeneity of effect sizes in the meta-analysis was determined through Cochran's Q and $I^2$ is reported. Additionally, we tested for the interaction between rs5985504-T and sex on log triglycerides adjusting for the same covariates as the main analysis.

To determine the correlation of the effect sizes of variants on total cholesterol in the chrXq23 locus and the effect sizes of these variants on CAD, T2D, and BMI, we performed analysis of chrXq23 variation on these three outcomes adjusting for age, age2, sex, genotyping array, and PCs in the UK Biobank, using the main effects model assuming X inactivation. We correlated effect sizes of total cholesterol–variants with effect sizes of variants on CAD, T2D, or BMI limiting to total cholesterol–variants with a MAF > 0.05 and $P < 0.05$.

**Expression quantitative trait analyses.** We downloaded the v7 SNP gene association results in tissue-specific files from the GTEx portal (https://gtexportal.org/home/datasets). We limited to six tissues that have been implicated in lipid biology or CHD (Adipose_Subcutaneous, Small_Intestine_Terminal_Ileum, Adipose_Visceral_Omentum, Whole_Blood, Liver, Muscle_Skeletal) and looked at expression of eight genes within the ChrXq23 region (ACSL4, TMEM164, AMMECR1, RTL9, CHRDL1, PAK3, CAPN6, DCX). We set our significance threshold to 0.001 (0.05/[6 tissues × 8 genes]). First, we determined eQTLs of our top association with lipids. Second, we performed correlation of our lipid–variant associations with the association of each of the eight genes expression on the variants using the gene transcripts ± 100 KB. Lastly, we used the lmekin function in the R package kinship2 to run linear mixed effects models and predict the lipid–variant test statistic (Z = beta/SE) from the expression–variant test statistic to adjust for the correlation between the variants.

**Lipid subfractions association analyses**. Concentrations of lipoprotein particles were measured at LipoScience, Inc. (Raleigh, NC) using NMR spectroscopy on plasma EDTA specimens. LipoScience has developed validated software for analysis of NMR measured LipoProfile spectra that uses an optimized deconvolution algorithm to quantify lipoprotein subspecies[66,67]. MESA was measured with the LipoProfile-III assay while FHS samples were measured with the LipoProfile-I assay, which provides less accuracy for some measurements but is similar to LipoProtein-III. We associated lipoprotein profiles with top associated SNPs within up to 1,802 FHS and 4551 MESA participants adjusting for age, sex, and lipid-lowering therapy.

For individuals who participated in ARIC study visit 4 (1996–1998), a homogeneous assay method was used for the direct measurement of sdLDL-C in plasma (sd-LDL-EX "Seiken", Denka Seiken, Tokyo, Japan) on a Hitachi 917 automated chemistry analyzer[68]. We associated top associated SNPs with ARIC participants adjusting for age, sex, lipid-lowering therapy, race, study center, and the first 11 principal components of ancestry.

**Rare variant association analyses**. We performed the SKAT test to associate aggregates of rare coding variants with blood lipid levels within TOPMed as implemented by GENESIS v2.14.4 in the CHARGE Analysis Commons[69,70]. For this gene-based test, high confidence loss-of-function (HC LOF by LOFTEE[71]) and missense metaSVM[72] damaging variants with MAF < 1% were collapsed into regions based on the gene annotations generated by snpEff 4.3t (http://snpeff.sourceforge.net/) using the GRCh38.86 database[73].

**Reporting summary**. Further information on research design is available in the Nature Research Reporting Summary linked to this article.

## Data availability

Controlled access of the individual-level TOPMed data is available through dbGaP, and the individual-level UK Biobank data are available upon application to the UK Biobank (https://www.ukbiobank.ac.uk/). FinnGen summary-level data are fully freely available at https://www.finngen.fi/en/access_results. Individual-level access to FinnGen and HUNT cohorts may be obtained through reasonable request and suitable institutional review board approvals. The dbGaP accessions for TOPMed cohorts are as follows: Atherosclerosis Risk in Communities (ARIC) phs001211 and phs000280; Old Order Amish phs000956 and phs000391; Mt Sinai BioMe Biobank phs001644 and phs000925; Coronary Artery Risk Development in Young Adults (CARDIA) phs001612 and phs000285; Cleveland Family Study (CFS) phs000954 and phs000284; Cardiovascular Health Study (CHS) phs001368; Diabetes Heart Study (DHS) phs001412 and phs001012; Framingham Heart Study (FHS) phs000974 and phs000007; Genetic Epidemiology Network of Arteriopathy (GENOA) phs001345 and phs001238; Genetics of Lipid-Lowering Drugs and Diet Network (GOLDN) phs001359 and phs000741; Genetic Epidemiology Network of Salt Sensitivity (GenSalt) phs001217 and phs000784; Genetic Studies of Atherosclerosis Risk (GeneSTAR) phs001218 and phs000375; Hispanic Community Health Study—Study of Latinos (HCHS/SOL) phs001395 and phs000810; Hypertension Genetic Epidemiology Network and Genetic Epidemiology Network of Arteriopathy (HyperGEN) phs001293; Jackson Heart Study (JHS) phs000964 and phs000286; Multi-Ethnic Study of Atherosclerosis (MESA) phs001416 and phs000209; Massachusetts General Hospital Atrial Fibrillation Study (MGH_AF) phs001062 and phs001001; San Antonio Family Study (SAFS) phs001215 and phs000462; Samoan Adiposity Study phs000972 and phs000914; Taiwan Study of Hypertension using Rare Variants (THRV) phs001387; Women's Health Initiative (WHI) phs001237 and phs000200. Source data are provided with this paper.

## Code availability

The variant calling software tools are under active development; updated versions can be accessed at http://github.com/atks/vt, http://github.com/hyunminkang/apigenome, and https://github.com/statgen/topmed_variant_calling.

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

## Acknowledgements

This work was supported by grants from the National Heart, Lung, and Blood Institute (NHLBI): R01HL142711 (to P.N. and G.M.P.), K08HL140203 (to P.N.), R03HL141439 and K01HL125751 (to G.M.P.). P.N. is also supported by a Hassenfeld Scholar Award from the Massachusetts General Hospital, Fondation Leducq (TNE-18CVD04), and additional grants from the National Heart, Lung, and Blood Institute (R01HL148565 and R01HL148050). P.S.de.V. is supported by American Heart Association grant number 18CDA34110116. B.E.C. and J.L. are supported by R35HL135818, HL113338, and HL46380. B.E.C. is also supported by K01HL135405. S.L. is supported by NIH grant 1R01HL139731 and American Heart Association 18SFRN34250007. Whole-genome sequencing (WGS) for the Trans-Omics in Precision Medicine (TOPMed) program was supported by the National Heart, Lung, and Blood Institute (NHLBI). Centralized read mapping and genotype calling, along with variant quality metrics and filtering were provided by the TOPMed Informatics Research Center (3R01HL-117626-02S1; contract HHSN268201800002I). Phenotype harmonization, data management, sample-identity QC, and general study coordination were provided by the TOPMed Data Coordinating Center (3R01HL-120393-02S1; contract HHSN268201800001I). We gratefully acknowledge the studies and participants who provided biological samples and data for TOPMed. Please refer to Supplementary Notes 1–4 for study-specific acknowledgements. The views expressed in this manuscript are those of the authors and do not necessarily represent the views of the National Heart, Lung, and Blood Institute; the National Institutes of Health; or the U.S. Department of Health and Human Services.

## Author contributions

Conceptualization of work and wrote original draft of manuscript: P.N. and G.M.P. Data curation and provided resources for study: P.N., A.Pampana, S.E.G., S.E.R., P.S.d.V., J.G.B., B.W., J.A.B., L.A.-F., K.A., M.C.H., J.P.P., M.A., S.A., T.L.A., C.M.B., L.F.B., J.C.B., B.E.C., R.D., H.D., L.S.E., Y.-J.H., A.T.K., L.A.L., J.L., R.N.L., L.W.M., G.M., J.-Y.M., J.R.O., N.D.P., J.M.P., A.M.S., M.T., F.F.W., D.E.W., L.R.Y., J.B., E.B., D.W.B., D.M.L.-J., Y.-C.C., Y.I.C., W.J.C., J.E.C., M.J.D., S.K.D., P.T.E., S.A.L., M.F., B.I.F., S.G., R.A.G., J.H., G.P.J., R.C.K., S.L.K., S.K., C.K., L.A.C., C.C.L., D.M.L., S.A.L., K.A.V.M., S.T.M., A.C.M., K.E.N., A.Palotie, C.J.P., B.M.P., D.C.R., S.Redline, A.P.R., D.S., J.-.S.S., R.S.V., J.I.R., S.S.R., S.Ripatti, and G.M.P. Statistical Analyses: P.N., A.Pampana, S.E.G., J.A.P., B.W., J.A.B., and G.M.P. Acquired funding and supervision of contributing studies: P.N., S.E.R., T.L.A., J.C.B., Y.-J.H., R.N.L., M.E.M., J.R.O., P.A.P., J.G.W., G.A., D.K.A., L.C.B., J.B., E.B., Y.-C.C., A.C., J.E.C., K.H., S.L.K., E.K., R.W.K., R.J.L., R.A.M., B.D.M., D.A.N., K.E.N., B.M.P., A.V.S., C.W., and G.M.P. Critically revised manuscript: P.N., A.Pampana, S.E.G., S.E.R., P.S.d.V., M.A., L.F.B., R.D., L.S.E., M.R.I., L.A.L., R.N.L., N.D.P., P.A.P., L.R.Y., J.G.W., L.C.B., J.B., A.C., J.E.C., M.F., S.L.K., E.K., R.J.L., R.A.M., A.C.M., K.E.N., R.P.T., and G.M.P. All authors approved the final manuscript.

## Competing interests

P.N. reports grants from Amgen, Apple, Boston Scientific, and Novartis, and consulting income from Apple, Blackstone Life Sciences, Genentech, and Novartis. S.L. reports grants from Bristol Myers Squibb / Pfizer, Bayer AG, and Boehringer Ingelheim, and consulting income from Bristol Myers Squibb / Pfizer and Bayer AG. P.E. is supported by a grant from Bayer AG to the Broad Institute focused on the genetics and therapeutics of

cardiovascular diseases. P.E. reports consulting income from Bayer AG, Quest Diagnostics, and Novartis. All others declare no competing interests for the present work.

**Additional information**

Pradeep Natarajan [1,2,3] ✉, Akhil Pampana [1,2], Sarah E. Graham [4], Sanni E. Ruotsalainen [5], James A. Perry [6], Paul S. de Vries [7], Jai G. Broome [8], James P. Pirruccello [1,2,3], Michael C. Honigberg [1,2,3], Krishna Aragam [1,2,3], Brooke Wolford [9], Jennifer A. Brody [10], Lucinda Antonacci-Fulton [11,12], Moscati Arden [13], Stella Aslibekyan [14], Themistocles L. Assimes [15,16], Christie M. Ballantyne [17,18], Lawrence F. Bielak [19], Joshua C. Bis [10], Brian E. Cade [20], Ron Do [13], Harsha Doddapaneni [21], Leslie S. Emery [8], Yi-Jen Hung [22], Marguerite R. Irvin [14], Alyna T. Khan [8], Leslie Lange [23], Jiwon Lee [20], Rozenn N. Lemaitre [10], Lisa W. Martin [24], Ginger Metcalf [21], May E. Montasser [6], Jee-Young Moon [25], Donna Muzny [21], Jeffrey R. O'Connell [6], Nicholette D. Palmer [26], Juan M. Peralta [27], Patricia A. Peyser [19], Adrienne M. Stilp [8], Michael Tsai [28], Fei Fei Wang [8], Daniel E. Weeks [29], Lisa R. Yanek [30], James G. Wilson [31,32], Goncalo Abecasis [33], Donna K. Arnett [34], Lewis C. Becker [30], John Blangero [27], Eric Boerwinkle [7,21], Donald W. Bowden [26], Yi-Cheng Chang [35], Yii-Der I. Chen [36], Won Jung Choi [37], Adolfo Correa [38], Joanne E. Curran [27], Mark J. Daly [2,5,39], Susan K. Dutcher [11,12], Patrick T. Ellinor [2,40], Myriam Fornage [41], Barry I. Freedman [42], Stacey Gabriel [43], Soren Germer [44], Richard A. Gibbs [21,45], Jiang He [46], Kristian Hveem [47,48], Gail P. Jarvik [49], Robert C. Kaplan [25,50], Sharon L. R. Kardia [19], Eimear Kenny [13], Ryan W. Kim [37], Charles Kooperberg [50], Cathy C. Laurie [8], Seonwook Lee [37], Don M. Lloyd-Jones [51], Ruth J. F. Loos [13,52], Steven A. Lubitz [40], Rasika A. Mathias [30], Karine A. Viaud Martinez [53], Stephen T. McGarvey [54], Braxton D. Mitchell [6,55], Deborah A. Nickerson [56,57], Kari E. North [58], Aarno Palotie [2,5,39], Cheol Joo Park [37], Bruce M. Psaty [10,59,60], D. C. Rao [61], Susan Redline [20], Alexander P. Reiner [50], Daekwan Seo [37], Jeong-Sun Seo [37], Albert V. Smith [33,62], Russell P. Tracy [63], Ramachandran S. Vasan [64,65,66], Sekar Kathiresan [2,3,67,68], L. Adrienne Cupples [66,69], Jerome I. Rotter [36], Alanna C. Morrison [7], Stephen S. Rich [70], Samuli Ripatti [2,5,71], Cristen Willer [4,9,72], NHLBI Trans-Omics for Precision Medicine (TOPMed) Consortium*, FinnGen* & Gina M. Peloso [69] ✉

[1]Cardiovascular Research Center, Massachusetts General Hospital, Boston, MA, USA. [2]Program in Medical and Population Genetics, Broad Institute, Cambridge, MA, USA. [3]Department of Medicine, Harvard Medical School, Boston, MA, USA. [4]Department of Internal Medicine: Cardiology, University of Michigan, Ann Arbor, MI, USA. [5]Institute for Molecular Medicine Finland, University of Helsinki, Helsinki, Finland. [6]University of Maryland School of Medicine, Division of Endocrinology, Diabetes and Nutrition and Program for Personalized and Genomic Medicine, Baltimore, MD, USA. [7]Human Genetics Center, Department of Epidemiology, Human Genetics, and Environmental Sciences, School of Public Health, The University of Texas Health Science Center at Houston, Houston, TX, USA. [8]Department of Biostatistics, University of Washington, Seattle, WA, USA. [9]Department of Computational Medicine and Bioinformatics, University of Michigan, Ann Arbor, MI, USA. [10]Cardiovascular Health Research Unit, Department of Medicine, University of Washington, Seattle, WA, USA. [11]The McDonnell Genome Institute, Washington University School of Medicine, St. Louis, MO, USA. [12]Department of Genetics, Washington University in St. Louis, St. Louis, MO, USA. [13]The Charles Bronfman Institute for Personalized Medicine, Ichan School of Medicine at Mount Sinai, New York, NY, USA. [14]Department of Epidemiology, School of Public Health, University of Alabama at Birmingham, Birmingham, AL, USA. [15]Department of Medicine, Stanford University

School of Medicine, Stanford, CA, USA. [16]VA Palo Alto Health Care System, Palo Alto, CA, USA. [17]Section of Cardiovascular Research, Baylor College of Medicine, Houston, TX, USA. [18]Houston Methodist Debakey Heart and Vascular Center, Houston, TX, USA. [19]Department of Epidemiology, School of Public Health, University of Michigan, Ann Arbor, MI, USA. [20]Department of Medicine, Brigham and Women's Hospital, Harvard Medical School, Boston, MA, USA. [21]Human Genome Sequencing Center, Baylor College of Medicine, Houston, TX, USA. [22]Division of Endocrine and Metabolism, Tri-Service General Hospital Songshan branch, Taipei, Taiwan. [23]Division of Biomedical Informatics and Personalized Medicine, Department of Medicine, University of Colorado Anschutz Medical Campus, Aurora, CO, USA. [24]Division of Cardiology, George Washington University School of Medicine and Healthcare Sciences, Washington, DC, USA. [25]Department of Epidemiology and Population Health, Albert Einstein College of Medicine, Bronx, NY, USA. [26]Department of Biochemistry, Wake Forest School of Medicine, Winston-Salem, NC, USA. [27]Department of Human Genetics and South Texas Diabetes and Obesity Institute, University of Texas Rio Grande Valley School of Medicine, Brownsville, TX, USA. [28]Department of Laboratory Medicine and Pathology, University of Minnesota, Minneapolis, MN, USA. [29]Departments of Human Genetics and Biostatistics, University of Pittsburgh, Pittsburgh, Pittsburgh, PA, USA. [30]Department of Medicine, Johns Hopkins University School of Medicine, Baltimore, MD, USA. [31]Division of Cardiovascular Medicine, Beth Israel Deaconess Medical Center, Boston, MA, USA. [32]Department of Physiology and Biophysics, University of Mississippi Medical Center, Jackson, MS, USA. [33]Department of Biostatistics, University of Michigan, Ann Arbor, MI, USA. [34]Deans office, School of Public Health, University of Kentucky, Lexington, KY, USA. [35]Department of Internal Medicine, National Taiwan University Hospital, Taipei, Taiwan. [36]The Institute for Translational Genomics and Population Sciences, Department of Pediatrics, The Lundquist Institute for Biomedical Innovation at Harbor-UCLA Medical Center, Torrance, CA, USA. [37]Psomagen. Inc. (formerly Macrogen USA), Rockville, MD, USA. [38]Department of Medicine, University of Mississippi Medical Center, Jackson, MS, USA. [39]Massachusetts General Hospital, Harvard Medical School, Boston, MA, USA. [40]Cardiac Arrhythmia Service and Cardiovascular Research Center Massachusetts General Hospital, Boston, MA, USA. [41]Institute of Molecular Medicine, the University of Texas Health Science Center at Houston, Houston, TX, USA. [42]Department of Internal Medicine, Section on Nephrology, Wake Forest School of Medicine, Winston-, Salem, NC, USA. [43]Genomics Platform, Broad Institute of Harvard and MIT, Cambridge, MA, USA. [44]New York Genome Center, New York, NY, USA. [45]Department of Molecular and Human Genetics, Baylor College of Medicine, Houston, TX, USA. [46]Department of Epidemiology, Tulane University School of Public Health and Tropical Medicine, and Tulane University Translational Science Institute, Tulane University, New Orleans, LA, USA. [47]Department of Public Health and General Practice, HUNT Research Centre, Norwegian University of Science and Technology, Levanger, Norway. [48]K. G. Jebsen Center for Genetic Epidemiology, Dept of Public Health and Nursing, Faculty of Medicine and Health Sciences, Norwegian University of Science and Technology (NTNU), Trondheim, Norway. [49]Departments of Medicine (Medical Genetics) and Genome Sciences, University of Washington, Seattle, WA, USA. [50]Division of Public Health Sciences, Fred Hutchinson Cancer Research Center, Seattle, WA, USA. [51]Preventive Medicine, Feinberg School of Medicine, Northwestern University, Chicago, IL, USA. [52]The Mindich Child Health and Development Institute, Ichan School of Medicine at Mount Sinai, New York, NY, USA. [53]Illumina Laboratory Services, Illumina inc., San Diego, CA, USA. [54]Department of Epidemiology and International Health Institute, Brown University, Providence, RI, USA. [55]Geriatrics Research and Education Clinical Center, Baltimore Veterans Administration Medical Center, Baltimore, MD, USA. [56]Department of Genome Sciences, University of Washington, Seattle, WA, USA. [57]University of Washington Center for Mendelian Genomics, Seattle, WA, USA. [58]Department of Epidemiology, Gillings School of Global Public Health, University of North Carolina at Chapel Hill, Chapel Hill, NC, USA. [59]Kaiser Permanente Washington Health Research Institute, Seattle, WA, USA. [60]Departments of Epidemiology and Health Services, University of Washington, Seattle, WA, USA. [61]Division of Biostatistics, Washington University School of Medicine, St. Louis, MO, USA. [62]The Icelandic Heart Association, Kopavogur, Iceland. [63]Departments of Pathology & Laboratory Medicine and Biochemistry, Larrner College of Medicine, University of Vermont, Colchester, VT, USA. [64]Sections of Preventive Medicine and Epidemiology and Cardiology, Department of Medicine, Boston University School of Medicine, Boston, MA, USA. [65]Department of Epidemiology, Boston University School of Public Health, Boston, MA, USA. [66]NHLBI Framingham Heart Study, Framingham, MA, USA. [67]Center for Genomic Medicine, Massachusetts General Hospital, Boston, MA, USA. [68]Verve Therapeutics, Cambridge, MA, USA. [69]Department of Biostatistics, Boston University School of Public Health, Boston, MA, USA. [70]Center for Public Health Genomics, University of Virginia, Charlottesville, VA, USA. [71]Department of Public Health, Faculty of Medicine, University of Helsinki, Helsinki, Finland. [72]Department of Human Genetics, University of Michigan, Ann Arbor, MI, USA. *Lists of authors and their affiliations appear at the end of the paper.
✉email: pnatarajan@mgh.harvard.edu; gpeloso@bu.edu

## NHLBI Trans-Omics for Precision Medicine (TOPMed) Consortium

Namiko Abe[44], Christine Albert[73], Laura Almasy[74], Alvaro Alonso[75], Seth Ament[76], Peter Anderson[77], Pramod Anugu[78], Deborah Applebaum-Bowden[79], Dan Arking[80], Allison Ashley-Koch[81], Paul Auer[82], Dimitrios Avramopoulos[80], John Barnard[83], Kathleen Barnes[84], R. Graham Barr[85], Emily Barron-Casella[80], Terri Beaty[80], Diane Becker[86], Rebecca Beer[87], Ferdouse Begum[80], Amber Beitelshees[76], Emelia Benjamin[88], Marcos Bezerra[89], Larry Bielak[90], Thomas Blackwell[90], Russell Bowler[91], Ulrich Broeckel[92], Karen Bunting[44], Esteban Burchard[93], Erin Buth[94], Jonathan Cardwell[84], Cara Carty[95], Richard Casaburi[96], James Casella[80], Mark Chaffin[97], Christy Chang[76], Daniel Chasman[98], Sameer Chavan[84], Bo-Juen Chen[44], Wei-Min Chen[99], Michael Cho[100], Seung Hoan Choi[97], Lee-Ming Chuang[101], Mina Chung[102], Matthew P. Conomos[103], Elaine Cornell[104], Carolyn Crandall[96], James Crapo[105], Jeffrey Curtis[90], Brian Custer[106], Coleen Damcott[76], Dawood Darbar[107], Sayantan Das[90], Sean David[108], Colleen Davis[77], Michelle Daya[84], Mariza de Andrade[109], Michael DeBaun[110], Ranjan Deka[111], Dawn DeMeo[100], Scott Devine[76], Qing Duan[112], Ravi Duggirala[113], Jon Peter Durda[104], Susan Dutcher[114], Charles Eaton[115], Lynette Ekunwe[78], Charles Farber[99], Leanna Farnam[100], Tasha Fingerlin[116], Matthew Flickinger[90], Nora Franceschini[117], Mao Fu[76], Stephanie M. Fullerton[77], Lucinda Fulton[114], Weiniu Gan[87], Yan Gao[78], Margery Gass[118], Bruce Gelb[119], Xiaoqi (Priscilla) Geng[90],

Soren Germer[44], Chris Gignoux[120], Mark Gladwin[121], David Glahn[122], Stephanie Gogarten[77], Da-Wei Gong[76], Harald Goring[123], C. Charles Gu[114], Yue Guan[76], Xiuqing Guo[124], Jeff Haessler[125], Michael Hall[78], Daniel Harris[76], Nicola Hawley[122], Ben Heavner[94], Susan Heckbert[77], Ryan Hernandez[126], David Herrington[127], Craig Hersh[128], Bertha Hidalgo[129], James Hixson[130], John Hokanson[84], Elliott Hong[76], Karin Hoth[131], Chao (Agnes) Hsiung[132], Haley Huston[133], Chii Min Hwu[134], Rebecca Jackson[135], Deepti Jain[77], Cashell Jaquish[87], Min A. Jhun[90], Jill Johnsen[136], Andrew Johnson[87], Craig Johnson[77], Rich Johnston[75], Kimberly Jones[80], Hyun Min Kang[137], Laura Kaufman[100], Shannon Kelly[106], Michael Kessler[76], Greg Kinney[84], Barbara Konkle[138], Holly Kramer[139], Stephanie Krauter[77], Christoph Lange[140], Ethan Lange[84], Cecelia Laurie[77], Meryl LeBoff[100], Seunggeun Shawn Lee[90], Wen-Jane Lee[134], Jonathon LeFaive[90], David Levine[77], Dan Levy[87], Joshua Lewis[76], Yun Li[112], Honghuang Lin[141], Keng Han Lin[90], Xihong Lin[142], Simin Liu[143], Yongmei Liu[144], Kathryn Lunetta[141], James Luo[87], Michael Mahaney[145], Barry Make[80], Ani Manichaikul[99], JoAnn Manson[100], Lauren Margolin[97], Susan Mathai[84], Patrick McArdle[76], Merry-Lynn McDonald[129], Sean McFarland[146], Caitlin McHugh[147], Hao Mei[78], Deborah A. Meyers[148], Julie Mikulla[87], Nancy Min[78], Mollie Minear[87], Ryan L. Minster[121], Solomon Musani[149], Stanford Mwasongwe[78], Josyf C. Mychaleckyj[99], Girish Nadkarni[119], Rakhi Naik[80], Take Naseri[150], Sergei Nekhai[151], Sarah C. Nelson[94], Deborah Nickerson[77], Jeff O'Connell[152], Tim O'Connor[76], Heather Ochs-Balcom[153], James Pankow[154], George Papanicolaou[87], Margaret Parker[100], Afshin Parsa[76], Sara Penchev[105], Marco Perez[120], Ulrike Peters[155], Lawrence S. Phillips[75], Sam Phillips[77], Toni Pollin[76], Wendy Post[156], Julia Powers Becker[157], Meher Preethi Boorgula[84], Michael Preuss[119], Dmitry Prokopenko[146], Pankaj Qasba[87], Dandi Qiao[100], Zhaohui Qin[75], Nicholas Rafaels[158], Laura Raffield[159], Laura Rasmussen-Torvik[160], Aakrosh Ratan[99], Robert Reed[76], Elizabeth Regan[105], Muagututi'a Sefuiva Reupena[161], Ken Rice[77], Dan Roden[162], Carolina Roselli[97], Ingo Ruczinski[80], Pamela Russell[84], Sarah Ruuska[163], Kathleen Ryan[76], Ester Cerdeira Sabino[164], Phuwanat Sakornsakolpat[100], Shabnam Salimi[76], Steven Salzberg[80], Kevin Sandow[165], Vijay G. Sankaran[166], Christopher Scheller[90], Ellen Schmidt[90], Karen Schwander[114], David Schwartz[84], Frank Sciurba[121], Christine Seidman[167], Jonathan Seidman[168], Vivien Sheehan[169], Amol Shetty[76], Aniket Shetty[84], Wayne Hui-Heng Sheu[134], M. Benjamin Shoemaker[170], Brian Silver[171], Edwin Silverman[100], Jennifer Smith[90], Josh Smith[77], Nicholas Smith[172], Tanja Smith[44], Sylvia Smoller[173], Beverly Snively[174], Tamar Sofer[100], Nona Sotoodehnia[77], Elizabeth Streeten[76], Jessica Lasky Su[175], Yun Ju Sung[114], Jody Sylvia[100], Adam Szpiro[77], Carole Sztalryd[76], Daniel Taliun[90], Hua Tang[176], Margaret Taub[80], Kent D. Taylor[177], Simeon Taylor[76], Marilyn Telen[81], Timothy A. Thornton[77], Lesley Tinker[95], David Tirschwell[77], Hemant Tiwari[129], Dhananjay Vaidya[80], Peter VandeHaar[90], Scott Vrieze[178], Tarik Walker[84], Robert Wallace[131], Avram Walts[84], Emily Wan[100], Heming Wang[179], Karol Watson[96], Bruce Weir[77], Scott Weiss[100], Lu-Chen Weng[180], Kayleen Williams[77], L. Keoki Williams[181], Carla Wilson[100], Quenna Wong[77], Huichun Xu[76], Ivana Yang[84], Rongze Yang[76], Norann Zaghloul[76], Maryam Zekavat[97], Yingze Zhang[182], Snow Xueyan Zhao[105], Wei Zhao[90], Degui Zhi[130], Xiang Zhou[90], Xiaofeng Zhu[183], Michael Zody[44] & Sebastian Zoellner[90]

[73]Brigham & Women's Hospital, Cedars Sinai, Boston, Massachusetts, USA. [74]Children's Hospital of Philadelphia, University of Pennsylvania, Philadelphia, PA, USA. [75]Emory University, Atlanta, GA, USA. [76]University of Maryland, Baltimore, MD, USA. [77]University of Washington, Seattle, WA, USA. [78]University of Mississippi, Jackson, MS, USA. [79]National Institutes of Health, Bethesda, MD, USA. [80]Johns Hopkins University, Baltimore, MD, USA. [81]Duke University, Durham, NC, USA. [82]University of Wisconsin Milwaukee, Milwaukee, WI, USA. [83]Cleveland Clinic, Cleveland, OH, USA. [84]University of Colorado at Denver, Denver, CO, USA. [85]Columbia University, New York, NY, USA. [86]Johns Hopkins University, Medicine, Baltimore, MD, USA. [87]National Heart, Lung, and Blood Institute, National Institutes of Health, Bethesda, MD, USA. [88]Boston University, Massachusetts General Hospital, Boston University School of Medicine, Boston, MA, USA. [89]Fundação de Hematologia e Hemoterapia de Pernambuco - Hemope, Recife, Brazil. [90]University of Michigan, Ann Arbor, MI, USA. [91]National Jewish Health, National Jewish Health, Denver, CO, USA. [92]Medical College of Wisconsin, Milwaukee, WI, USA. [93]University of California, San Francisco, San Francisco, CA, USA. [94]University of Washington, Biostatistics, Seattle, WA, USA. [95]Women's Health Initiative, Seattle, WA, USA. [96]University of California, Los Angeles, Los Angeles, CA, USA. [97]Broad Institute, Cambridge, MA, USA. [98]Brigham & Women's Hospital, Division of Preventive Medicine, Boston, MA, USA. [99]University of Virginia, Charlottesville, VA, USA. [100]Brigham & Women's Hospital, Boston, MA, USA. [101]National Taiwan University, National

Taiwan University Hospital, Taipei, Taiwan. [102]Cleveland Clinic, Cleveland Clinic, Cleveland, OH, USA. [103]University of Washington, Biostatistics, Seattle, WA, USA. [104]University of Vermont, Burlington, VT, USA. [105]National Jewish Health, Denver, CO, USA. [106]Vitalant Research Institute, San Francisco, CA, USA. [107]University of Illinois at Chicago, Chicago, IL, USA. [108]University of Chicago, Chicago, IL, USA. [109]Mayo Clinic, Health Sciences Research, Rochester, MN, USA. [110]Vanderbilt University, Nashville, TN, USA. [111]University of Cincinnati, Cincinnati, OH, USA. [112]University of North Carolina, Chapel Hill, NC, USA. [113]University of Texas Rio Grande Valley School of Medicine, Edinburg, TX, USA. [114]Washington University in St Louis, St Louis, MO, USA. [115]Brown University, Providence, RI, USA. [116]National Jewish Health, Center for Genes, Environment and Health, Denver, CO, USA. [117]University of North Carolina, Epidemiology, Chapel Hill, NC, USA. [118]Fred Hutchinson Cancer Research Center, Seattle, WA, USA. [119]Icahn School of Medicine at Mount Sinai, New York, NY, USA. [120]Stanford University, Stanford, CA, USA. [121]University of Pittsburgh, Pittsburgh, PA, USA. [122]Yale University, New Haven, CT, USA. [123]University of Texas Rio Grande Valley School of Medicine, San Antonio, TX, USA. [124]Lundquist Institute, Los Angeles, CA, USA. [125]Fred Hutchinson Cancer Research Center, Women's Health Initiative, Seattle, WA, USA. [126]McGill University, University of California, San Francisco, CA, USA. [127]Wake Forest Baptist Health, Winston-Salem, NC, USA. [128]Brigham & Women's Hospital, Channing Division of Network Medicine, Boston, MA, USA. [129]University of Alabama, Birmingham, AL, USA. [130]University of Texas Health at Houston, Houston, TX, USA. [131]University of Iowa, Iowa City, IA, USA. [132]National Health Research Institute Taiwan, Institute of Population Health Sciences, NHRI, Miaoli County, TW, Taiwan. [133]Blood Works Northwest, Seattle, WA, USA. [134]Taichung Veterans General Hospital Taiwan, Taichung City, Taiwan. [135]Ohio State University Wexner Medical Center, Internal Medicine, DIvision of Endocrinology, Diabetes and Metabolism, Columbus, OH, USA. [136]Blood Works Northwest, University of Washington, Seattle, WA, USA. [137]University of Michigan, Biostatistics, Ann Arbor, MI, USA. [138]Blood Works Northwest, Seattle, WA, USA. [139]Loyola University, Public Health Sciences, Maywood, IL, USA. [140]Harvard School of Public Health, Biostats, Boston, MA, USA. [141]Boston University, Boston, MA, USA. [142]Harvard School of Public Health, Boston, MA, USA. [143]Brown University, Women's Health Initiative, Epidemiology, Providence, RI, USA. [144]Duke University, Cardiology, Durham, NC, USA. [145]University of Texas Rio Grande Valley School of Medicine, Brownsville, TX, USA. [146]Harvard University, Cambridge, MA, USA. [147]University of Washington, Biostatistics, Seattle, WA, USA. [148]University of Arizona, Tucson, AZ, USA. [149]University of Mississippi, Medicine, Jackson, MP, USA. [150]Ministry of Health, Government of Samoa, Apia, WS, Samoa. [151]Howard University, Washington, DC, USA. [152]University of Maryland, Balitmore, MD, USA. [153]University at Buffalo, Buffalo, NY, USA. [154]University of Minnesota, Minneapolis, MN, USA. [155]Fred Hutchinson Cancer Research Center, University of Washington, Seattle, WA, USA. [156]Johns Hopkins University, Cardiology/Medicine, Baltimore, MD, USA. [157]University of Colorado at Denver, Medicine, Denver, CO, USA. [158]University of Colorado at Denver, Denver, CO, USA. [159]University of North Carolina, Genetics, Chapel Hill, NC, USA. [160]Northwestern University, Chicago, IL, USA. [161]Lutia I Puava Ae Mapu I Fagalele, Apia, WS, Samoa. [162]Vanderbilt University, Medicine, Pharmacology, Biomedicla Informatics, Nashville, TN, USA. [163]Blood Works Northwest, Seattle, WA, USA. [164]Universidade de Sao Paulo, Faculdade de Medicina, Sao Paulo, Brazil. [165]Lundquist Institute, TGPS, Torrance, CA, USA. [166]Broad Institute, Harvard University, Division of Hematology/Oncology, Boston, MA, USA. [167]Harvard Medical School, Genetics, Boston, MA, USA. [168]Harvard Medical School, Boston, MA, USA. [169]Baylor College of Medicine, Pediatrics, Houston, TX, USA. [170]Vanderbilt University, Medicine/Cardiology, Nashville, TN, USA. [171]UMass Memorial Medical Center, Worcester, MA, USA. [172]University of Washington, Epidemiology, Seattle, WA, USA. [173]Albert Einstein College of Medicine, New York, NY, USA. [174]Wake Forest Baptist Health, Biostatistical Sciences, Winston-Salem, NC, USA. [175]Brigham & Women's Hospital, Boston, MA, USA. [176]Stanford University, Genetics, Stanford, CA, USA. [177]Lundquist Institute, Institute for Translational Genomics and Populations Sciences, Torrance, CA, USA. [178]University of Colorado at Boulder, University of Minnesota, Boulder, CO, USA. [179]Brigham & Women's Hospital, Partners.org, Boston, MA, USA. [180]Massachusetts General Hospital, Boston, MA, USA. [181]Henry Ford Health System, Detroit, MI, USA. [182]University of Pittsburgh, Medicine, Pittsburgh, PA, USA. [183]Department of Population and Quantitative Health Sciences, Case Western Reserve University, Cleveland, OH, USA.

## FinnGen

Aarno Palotie[184], Mark Daly[184], Howard Jacob[185], Athena Matakidou[186], Heiko Runz[187], Sally John[187], Robert Plenge[188], Mark McCarthy[189], Julie Hunkapiller[189], Meg Ehm[190], Dawn Waterworth[190], Caroline Fox[191], Anders Malarstig[192], Kathy Klinger[193], Kathy Call[193], Tomi Mkel[194], Jaakko Kaprio[195], Petri Virolainen[196], Kari Pulkki[196], Terhi Kilpi[197], Markus Perola[197], Jukka Partanen[198], Anne Pitkranta[199], Riitta Kaarteenaho[200], Seppo Vainio[200], Kimmo Savinainen[201], Veli-Matti Kosma[202], Urho Kujala[203], Outi Tuovila[204], Minna Hendolin[204], Raimo Pakkanen[204], Jeff Waring[185], Bridget Riley-Gillis[185], Jimmy Liu[187], Shameek Biswas[188], Dorothee Diogo[191], Catherine Marshall[192], Xinli Hu[192], Matthias Gossel[193], Samuli Ripatti[184], Johanna Schleutker[196], Mikko Arvas[198], Olli Carpen[199], Reetta Hinttala[200], Johannes Kettunen[200], Reijo Laaksonen[201], Arto Mannermaa[202], Juha Paloneva[203], Hilkka Soininen[205], Valtteri Julkunen[205], Anne Remes[206], Reetta Klviinen[205], Mikko Hiltunen[205], Jukka Peltola[207], Pentti Tienari[199], Juha Rinne[208], Adam Ziemann[185], Jeffrey Waring[185], Sahar Esmaeeli[185], Nizar Smaoui[185], Anne Lehtonen[185], Susan Eaton[187], Sanni Lahdenper[187], John Michon[189], Geoff Kerchner[189], Natalie Bowers[189], Edmond Teng[189], John Eicher[191], Vinay Mehta[191], Padhraig Gormley[191], Kari Linden[192], Christopher Whelan[192], Fanli Xu[190], David Pulford[190], Martti Frkkil[199], Sampsa Pikkarainen[199], Airi Jussila[207], Timo Blomster[206], Mikko Kiviniemi[205], Markku Voutilainen[208], Bob Georgantas[185], Graham Heap[185], Fedik Rahimov[185], Keith Usiskin[188], Joseph Maranville[188], Tim Lu[189], Danny Oh[189], Kirsi Kalpala[192], Melissa Miller[192], Linda McCarthy[190], Kari Eklund[199], Antti Palomki[208], Pia Isomki[207], Laura Piril[208], Oili Kaipiainen-Seppnen[205], Johanna Huhtakangas[206], Apinya Lertratanakul[185], David Close[186], Marla Hochfeld[188], Nan Bing[192],

Jorge Esparza Gordillo[190], Nina Mars[209], Tarja Laitinen[207], Margit Pelkonen[205], Paula Kauppi[199], Hannu Kankaanranta[207], Terttu Harju[206], Steven Greenberg[188], Hubert Chen[189], Jo Betts[190], Soumitra Ghosh[190], Veikko Salomaa[210], Teemu Niiranen[210], Markus Juonala[208], Kaj Metsrinne[208], Mika Khnen[207], Juhani Junttila[206], Markku Laakso[205], Jussi Pihlajamki[205], Juha Sinisalo[199], Marja-Riitta Taskinen[199], Tiinamaija Tuomi[199], Jari Laukkanen[211], Ben Challis[186], Andrew Peterson[189], Audrey Chu[191], Jaakko Parkkinen[192], Anthony Muslin[193], Heikki Joensuu[199], Tuomo Meretoja[199], Lauri Aaltonen[199], Annika Auranen[212], Peeter Karihtala[206], Saila Kauppila[206], Pivi Auvinen[205], Klaus Elenius[208], Relja Popovic[185], Jennifer Schutzman[189], Andrey Loboda[191], Aparna Chhibber[191], Heli Lehtonen[192], Stefan McDonough[192], Marika Crohns[193], Diptee Kulkarni[190], Kai Kaarniranta[205], Joni Turunen[199], Terhi Ollila[199], Sanna Seitsonen[199], Hannu Uusitalo[207], Vesa Aaltonen[208], Hannele Uusitalo-Jrvinen[207], Marja Luodonp[206], Nina Hautala[206], Erich Strauss[189], Hao Chen[189], Anna Podgornaia[191], Joshua Hoffman[190], Kaisa Tasanen[206], Laura Huilaja[206], Katariina Hannula-Jouppi[199], Teea Salmi[207], Sirkku Peltonen[208], Leena Koulu[208], Ilkka Harvima[205], Ying Wu[192], David Choy[189], Anu Jalanko[184], Risto Kajanne[184], Ulrike Lyhs[184], Mari Kaunisto[184], Justin Wade Davis[185], Danjuma Quarless[185], Slav Petrovski[186], Chia-Yen Chen[187], Paola Bronson[187], Robert Yang[188], Diana Chang[189], Tushar Bhangale[189], Emily Holzinger[191], Xulong Wang[191], Xing Chen[192], sa Hedman[192], Kirsi Auro[190], Clarence Wang[193], Ethan Xu[193], Franck Auge[193], Clement Chatelain[193], Mitja Kurki[213], Juha Karjalainen[213], Aki Havulinna[184], Kimmo Palin[214], Priit Palta[184], Pietro Della Briotta Parolo[184], Wei Zhou[215], Susanna Lemmel[184], Manuel Rivas[216], Jarmo Harju[184], Arto Lehisto[184], Andrea Ganna[184], Vincent Llorens[184], Antti Karlsson[196], Kati Kristiansson[197], Kati Hyvrinen[198], Jarmo Ritari[198], Tiina Wahlfors[198], Miika Koskinen[217], Katri Pylks[200], Marita Kalaoja[200], Minna Karjalainen[200], Tuomo Mantere[200], Eeva Kangasniemi[201], Sami Heikkinen[202], Eija Laakkonen[203], Juha Kononen[203], Anu Loukola[199], Pivi Laiho[197], Tuuli Sistonen[197], Essi Kaiharju[197], Markku Laukkanen[197], Elina Jrvensivu[197], Sini Lhteenmki[197], Lotta Mnnikk[197], Regis Wong[197], Hannele Mattsson[197], Tero Hiekkalinna[197], Manuel Gonzlez Jimnez[197], Kati Donner[184], Kalle Prn[184], Javier Nunez-Fontarnau[184], Elina Kilpelinen[184], Timo P. Sipil[184], Georg Brein[184], Alexander Dada[184], Ghazal Awaisa[184], Anastasia Shcherban[184], Tuomas Sipil[184], Hannele Laivuori[184], Tuomo Kiiskinen[184], Harri Siirtola[218], Javier Gracia Tabuenca[218], Lila Kallio[219], Sirpa Soini[220], Kimmo Pitknen[221] & Teijo Kuopio[222]

[184]Institute for Molecular Medicine Finland, HiLIFE, University of Helsinki, Helsinki, Finland. [185]Abbvie, Chicago, IL, USA. [186]Astra Zeneca, Cambridge, UK. [187]Biogen, Cambridge, MA, USA. [188]Celgene, Summit, NJ, USA. [189]Genentech, San Francisco, CA, USA. [190]GlaxoSmithKline, Brentford, UK. [191]Merck, Kenilworth, NJ, USA. [192]Pfizer, New York, NY, USA. [193]Sanofi, Paris, France. [194]HiLIFE, University of Helsinki, Helsinki, Finland. [195]Institute for Molecular Medicine Finland, HiLIFE, Helsinki, Finland. [196]Auria Biobank / Univ. of Turku / Hospital District of Southwest Finland, Turku, Finland. [197]THL Biobank / The National Institute of Health and Welfare Helsinki, Helsinki, Finland. [198]Finnish Red Cross Blood Service / Finnish Hematology Registry and Clinical Biobank, Helsinki, Finland. [199]Hospital District of Helsinki and Uusimaa, Helsinki, Finland. [200]Northern Finland Biobank Borealis / University of Oulu / Northern Ostrobothnia Hospital District, Oulu, Finland. [201]Finnish Clinical Biobank Tampere / University of Tampere / Pirkanmaa Hospital District, Tampere, Finland. [202]Biobank of Eastern Finland / University of Eastern Finland / Northern Savo Hospital District, Kuopio, Finland. [203]Central Finland Biobank / University of Jyvskyl / Central Finland Health Care District, Jyvskyl, Finland. [204]Business Finland, Helsinki, Finland. [205]Northern Savo Hospital District, Kuopio, Finland. [206]Northern Ostrobothnia Hospital District, Oulu, Finland. [207]Pirkanmaa Hospital District, Tampere, Finland. [208]Hospital District of Southwest Finland, Turku, Finland. [209]Institute for Molecular Medicine Finland, HiLIFE, Helsinki, Finland. [210]The National Institute of Health and Welfare Helsinki, Helsinki, Finland. [211]Central Finland Health Care District, Jyvskyl, Finland. [212]Pirkanmaa Hospital District, Tampere, Finland. [213]Institute for Molecular Medicine Finland, HiLIFE, University of Helsinki, Finland / Broad Institute, Cambridge, MA, USA. [214]University of Helsinki, Helsinki, Finland. [215]Broad Institute, Cambridge, MA, USA. [216]University of Stanford, Stanford, CA, USA. [217]Hospital District of Helsinki and Uusimaa, Finland BB/HUS/Univ Hosp Districts, Helsinki, Finland. [218]University of Tampere, Tampere, Finland. [219]Auria Biobank, Turku, Finland. [220]THL Biobank, Helsinki, Finland. [221]Helsinki Biobank, Helsinki, Finland. [222]Central Finland Biobank, Jyvskyl, Finland.

