## [Peer Review File · Nature Communications]

Reviewer #3 (Remarks to the Author):

This report adds useful confirmatory and fine-mapping information to previously published reports of a cardiometabolic locus located on chromosome X.

Introduction: "Although the X chromosome comprises 5% of the genome, it is often ignored in genome-wide association analyses, including those for blood lipids.¹⁰": The reports of lipid and coronary disease associations with rs5943057, rs5985471, and rs5942937 are from germane GWAS that did not ignore the X-chromosome, please revise the introduction with citations.

Results: "To determine whether our signal was independent of previously reported variants in the region, we performed conditional analysis for the associated between total cholesterol and rs5942634 with rs5943057, rs5985471, and rs5942937 (Supplementary Table 11)."

Add citations when these variants are reported in the results. Revise the first paragraph of the discussion to frame your results in context of these "previously reported variants to cardiometabolic traits in the region".

Reviewer #4 (Remarks to the Author):

Based on my previous comments on an earlier version of the manuscript, the authors revised their manuscript in a number of important aspects. However, major concerns specifically with regard to the description of the methodology still remain. As a general line, much more detailed or structured explanations of the statistical models are required, and examples are again given in the following, partly still pertaining to my previous comments:

1. The authors now state that decisions on covariables in the different models were made by the study investigators, and this applies to numerous analyses reported. Still, this is not a statistical justification for the models. For instance, while it might make sense that different numbers of PCs are considered in different study groups (if justified based on the local population structure), it is not clear to me why, for example, sometimes age plus age² are included and sometimes only age. Also, it seems that within TOPMed, residuals were computed for lipid values using PCs and other covariables in the regression model; for later association analyses with SNPs, again PCs were included as covariables, for which the rationale is unclear to me. Further explanations are required, for example, concerning the coding of males in the genetic association analyses: Why was only the 0-2 coding chosen that assumes complete inactivation of one X chromosome in the females? Please note that these are only examples for the lack of detail in the description of the statistical methodology.

2. In their revision, the authors added details on the adjustment for multiple testing. This raises a question about the overall strategy of the association analysis: Generally, one of two options are sensible. First, there could be a discovery analysis and a replication analysis, in which case the adjustment would have to be in the replication analysis for the number of variants tested. In that case, a meta-analysis of both parts would, from a statistical point of view, not really be meaningful, but might be added purely descriptively. Second, the main analysis could be the meta-analysis (of the three studies), in which case the adjustment would have to be for the tests carried out there. Then, it would only be confusing to term the different parts "discovery" and "replication", and the tests in the different studies would be analysed purely descriptively.

3. For "replication", the authors sometimes state that $p < 0.05$ is used as a threshold in both UKBB and HUNT, but in other parts of the manuscript report that $p < 0.05$ is used only in UKBB and "evidence for associations in HUNT".

4. Concerning the meta-analysis of association studies on lipid levels, estimates for heterogeneity should always be provided. Also, the interpretation of effects requires some clarification: It seems that in TOPMed, lipid residuals from regression models were used for the association analyses, and that the association analyses themselves also included a number of covariables; in UKBB and HUNT, it is reported that inversely normalized lipids were used for the association analyses, with additional covariables. This implies that the dependent variable in the different studies has a different meaning, which raises the question of how to interpret the beta estimates in the three single studies and in the meta-analysis.

5. The adjustment for multiple testing within the PheWAS is described in the manuscript. Here, the authors need to be clear about the nature of their analyses, given that the PheWAS is explicitly described to be exploratory, which implies that no inferential statistics are performed. In addition, there needs to be an explanation on why exactly 80 traits are being investigated.

6. To illustrate a possible heterogeneity, the authors included p-values for heterogeneity in supplementary tables 9 and 10. There needs to be a description which test was being performed, and the p-values should be supplemented by estimates of heterogeneity.

7. The authors added estimates of the inflation factor lambda to the QQ plots. Given that these range between 1.09 and 1.17, it should be justified why these are interpreted to be negligible instead of accounting for this degree of inflation in the analyses.

8. The manuscript states: "In HUNT, UK Biobank, and FinnGen (Supplementary Table 12), we observed that the top lipid-lowering alleles at this locus were associated with reduced risk for CHD (Figure 2)." and "Notably, we observed consistently lower odds of T2D in both HUNT and FinnGen yielding, for rs5942648, a meta-analyzed OR = 0.97 (95% CI 0.96, 0.99; P = 1.4x10⁻⁵)". These interpretations are overstatements, because for CHD, this is only true for the meta-analysis but not for the single studies, where the confidence interval includes the 1 for both UKBB and FinnGen. Similarly, for T2D, this is only true for the meta-analysis and the point estimates, but the confidence interval for HUNT again includes 1.

Reviewer #5 (Remarks to the Author):

The authors present their data on an X-chromosome locus that associates with plasma lipid levels. Also provided are responses to a rigorous review of a previous version of the manuscript, for which the authors have tackled most issues. They have provided a very strong response and should be commended for doing so. On this basis many of the large issues have been dealt with but I have a few comments as detailed below.

A test for heterogeneity is not a substitution for an interaction test. In particular, heterogeneity estimated using Cochran's Q between only 2 groups is fairly meaningless and yet still highly significant for triglycerides suggesting that a formal interaction test would confirm that the effects are not similar between men and women. Please formally test this interaction and alter the text accordingly if a significant interaction is found.

Colocalisation analyses could be extended to include BMI, T2D and CAD (rather than gene expression) to avoid over interpreting a single variant result from a PHEWAS that ignores LD. Given the established relationships at HMGCR and PCSK9 with lipids, CAD but not consistently with T2D and BMI this relationship could also be investigated for your variant through appropriate mediation analyses. This relationship with cardiometabolic traits is not clear enough to justify the title of the paper.

Why, given there is no heterogeneity between ancestry in TOPMED did you only look at white-British in UK Biobank?

The methods for conditional analyses and how you estimate heterogeneity appear to be missing in the methods.

In conditional analyses you highlight your signal as only being partially explained by previously identified variants. i.e. still independent. However, given that the signal disappears for the published variants when you add your new variant I consider this new finding to be a refinement of the known loci rather than novel.

Response to Reviews

RE: ChrXq23 is associated with lower atherogenic lipid concentrations and favorable cardiometabolic indices

Reviewer #3:

This report adds useful confirmatory and fine-mapping information to previously published reports of a cardiometabolic locus located on chromosome X.

Author Response:

We thank the Reviewer for their input, which has improved the manuscript.

1. Introduction: “Although the X chromosome comprises 5% of the genome, it is often ignored in genome-wide association analyses, including those for blood lipids.¹⁰”: The reports of lipid and coronary disease associations with rs5943057, rs5985471, and rs5942937 are from germane GWAS that did not ignore the X-chromosome, please revise the introduction with citations.

Author Response:

We have revised the Introduction and add the citations to the text.

Manuscript Change:

“Although the X chromosome comprises 5% of the genome, it has only been studied in a few genome-wide association analyses for blood lipids and coronary disease.¹⁰⁻¹³”

2. Results: “To determine whether our signal was independent of previously reported variants in the region, we performed conditional analysis for the associated between total cholesterol and rs5942634 with rs5943057, rs5985471, and rs5942937 (Supplementary Table 11).”

Add citations when these variants are reported in the results. Revise the first paragraph of the discussion to frame your results in context of these “previously reported variants to cardiometabolic traits in the region”.

Author Response:

We have added the citations to the Results on line 330 when reporting the previously associated SNPs.

We revised the first sentence of the Discussion to frame the results in the context of the previous association in the region:

Manuscript Change:

“Using deep-coverage next-generation sequencing of the X chromosome in the NHLBI TOPMed program, we identified a locus that was previously associated

with lipids in primarily European ancestry datasets, which we now extend to diverse ancestries along with strong replication in two independent studies. In addition to replicating the associations of chrXq23 lipid-lowering alleles with reduced odds for CHD, we also observe associations with reduced T2D odds and favorable adiposity indices. These observations allow us to draw several conclusions about X chromosome genetic variation with blood lipid levels, as well as related cardiometabolic effects.”

Reviewer #4:

Based on my previous comments on an earlier version of the manuscript, the authors revised their manuscript in a number of important aspects. However, major concerns specifically with regard to the description of the methodology still remain. As a general line, much more detailed or structured explanations of the statistical models are required, and examples are again given in the following, partly still pertaining to my previous comments:

Author Response:

We appreciate the constructive feedback and address residual concerns below.

1. The authors now state that decisions on covariables in the different models were made by the study investigators, and this applies to numerous analyses reported. Still, this is not a statistical justification for the models. For instance, while it might make sense that different numbers of PCs are considered in different study groups (if justified based on the local population structure), it is not clear to me why, for example, sometimes age plus age² are included and sometimes only age. Also, it seems that within TOPMed, residuals were computed for lipid values using PCs and other covariables in the regression model; for later association analyses with SNPs, again PCs were included as covariables, for which the rationale is unclear to me. Further explanations are required, for example, concerning the coding of males in the genetic association analyses: Why was only the 0-2 coding chosen that assumes complete inactivation of one X chromosome in the females? Please note that these are only examples for the lack of detail in the description of the statistical methodology.

Author Response:

Age-squared is often included in genetic models of blood lipid levels as levels tend to increase as one ages and then decrease at older ages creating a parabolic shape to the relationship (Kathiresan S et al BMC Med Genet 2007; Demirkan A et al Eur J Hum Genet 2011; Varga T et al PLoS Genet 2014; Woo JG et al JLR 2014; Lu X et al Circ: Genom Precis Med 2015; Surakka I et al Nat Genet 2015; Hoffman TJ et al Nat Genet 2018). We have updated the lipid associations so that they all include the age-squared term for consistency. This has not changed the results. Additionally, we have made updates throughout the methods section to clarify the statistical models.

We used the 0-2 coding to model complete inactivation of one X chromosome in females as this was the default model and assumes random X inactivation since we don't know what allele is being expressed in women. The following clarification was added to page 32:

Manuscript Change:

“This modeling is consistent with random X inactivation of 1 of 2 X chromosomes in females yet consistent expression of the single X chromosome in males.

For SNPs with suggest evidence of association in TOPMed ($P < 1 \times 10^{-6}$), we sought replication in UKBB. For replication in UKBB unrelated individuals, we performed linear regression associations using R. Covariates included age, age², sex, the first 10 PCs in the model, and genotyping array.

For replication in HUNT, a cohort within a founder population, plasma lipids were analyzed using efficient linear mixed models implemented by BOLT-LMM v2.3.1 with covariates for sex, age, age², batch, and principle components 1-4. CAD was analyzed using SAIGE with birth year, batch, sex, and principle components 1-4 as covariates.”

2. In their revision, the authors added details on the adjustment for multiple testing. This raises a question about the overall strategy of the association analysis: Generally, one of two options are sensible. First, there could be a discovery analysis and a replication analysis, in which case the adjustment would have to be in the replication analysis for the number of variants tested. In that case, a meta-analysis of both parts would, from a statistical point of view, not really be meaningful, but might be added purely descriptively. Second, the main analysis could be the meta-analysis (of the three studies), in which case the adjustment would have to be for the tests carried out there. Then, it would only be confusing to term the different parts “discovery” and “replication”, and the tests in the different studies would be analysed purely descriptively.

Author Response:

We had used a more stringent threshold for significance than the Reviewer suggests. We have modified the adjustment for multiple testing to reflect a discovery and replication approach, as we do not have access to the full summary statistics from all the replication cohorts. We found 21 loci in our TOPMed analyses that met suggestive evidence of association ($P < 1 \times 10^{-6}$) that we brought forward for replication. We now apply a Bonferroni correction of 21 loci ($0.05/21=0.002$) in the replication cohorts. This does not change the results presented. We keep the meta-analyzed results for a descriptive nature. These results provide evidence that the associations are stronger than the TOPMed results alone.

Manuscript Change:

Results:

“Three variants showed evidence of replication ($P < 0.05/21 = 0.002$) in UK Biobank and in HUNT and additionally met a stringent threshold for statistical significance in the meta-analysis ($\alpha = 0.05/2.2\text{M variants}/4\text{ traits} = 5.7 \times 10^{-9}$) (**Table 1**).”

Methods:

“We took loci reaching a suggestive association with lipid levels ($P < 1 \times 10^{-6}$) in TOPMed onto replication in UKBB and HUNT. For replication, we used a significance level of 0.002 (Bonferroni correction for 21 loci) that met a suggestive level of association in TOPMed.”

Table 1 legend:

“Variants attaining $P < 1 \times 10^{-6}$ in TOPMed (discovery) and $P < 0.002$ in UK Biobank and HUNT (replication studies).”

3. For “replication”, the authors sometimes state that $p < 0.05$ is used as a threshold in both UKBB and HUNT, but in other parts of the manuscript report that $p < 0.05$ is used only in UKBB and “evidence for associations in HUNT”.

Author Response:

We have clarified the replication approach as outlined in response to point number 2, which also addresses this concern.

4. Concerning the meta-analysis of association studies on lipid levels, estimates for heterogeneity should always be provided. Also, the interpretation of effects requires some clarification: It seems that in TOPMed, lipid residuals from regression models were used for the association analyses, and that the association analyses themselves also included a number of covariables; in UKBB and HUNT, it is reported that inversely normalized lipids were used for the association analyses, with additional covariables. This implies that the dependent variable in the different studies has a different meaning, which raises the question of how to interpret the beta estimates in the three single studies and in the meta-analysis.

Author Response:

Estimates of heterogeneity (I^2) have now been added to **Table 1**. Additionally, we confirmed and clarified in the methods that all effect sizes from discovery and replication cohorts for lipid levels are on the mg/dl or log(mg/dl) for triglycerides. Our discovery inverse-normalized residuals were scaled by the trait standard deviations, as stated in the methods and the replication cohorts modeled the lipids in mg/dl units. This has been clarified in the Methods.

Manuscript Change:

In the methods on page 28:

“Effect sizes are reported in mg/dl or log(mg/dl) for TG.”

5. The adjustment for multiple testing within the PheWAS is described in the manuscript. Here, the authors need to be clear about the nature of their analyses,

given that the PheWAS is explicitly described to be exploratory, which implies that no inferential statistics are performed. In addition, there needs to be an explanation on why exactly 80 traits are being investigated.

Author Response:

We manually curated a set of 80 common disorders to perform PheWAS. We have clarified the text around the PheWAS in the Results (lines 353-355) to indicate that we are evaluating which phenotypes are associated with the chrXq23 locus:

Manuscript Change:

“To explore the range of phenotypes associated with the chrXq23 locus, we evaluated the associations of each of these three variants with 80 manually curated diverse clinical traits and conditions in the UK Biobank (**Supplementary Table 13**).”

And, in the Methods:

“A total of 80 manually curated traits were classified according to a combination of self-report and billing codes...”

6. To illustrate a possible heterogeneity, the authors included p-values for heterogeneity in supplementary tables 9 and 10. There needs to be a description which test was being performed, and the p-values should be supplemented by estimates of heterogeneity.

Author Response:

We have added the test of heterogeneity to the methods and supplemented all the heterogeneity p-values with I^2 estimates in Table 1 and in Supplementary Tables 9 and 10.

Manuscript Change:

“Heterogeneity of effect sizes in the meta-analysis was determined through Cochran's Q and I^2 is reported.”

7. The authors added estimates of the inflation factor lambda to the QQ plots. Given that these range between 1.09 and 1.17, it should be justified why these are interpreted to be negligible instead of accounting for this degree of inflation in the analyses.

Author Response:

We removed the subjective interpretation of the genomic control inflation factors. Given our discovery and replication study design, adjusting for the lambda values in our discovery analysis would just increase the threshold we are using to take SNPs forward for replication and our variants replicated in the additional cohorts. Importantly, our lead associated SNPs would still pass our significance threshold for taking them forward to replication after adjusting for the GC lambda.

SNP	Trait	beta	se	p	Chi-Sq	GC	GC corrected	Adjusted
-----	-------	------	----	---	--------	----	--------------	----------

						lambda	chi-sq	P-value
rs5942634	TC	-1.95	0.237	2.0x10-16	67.70	1.17	57.86	2.81E-14
rs5985504	log(TG)	0.019	0.0029	4.16x10-11	42.03	1.09	38.56	5.32E-10
rs5942648	LDL-C	-1.53	0.215	1.19x10-12	50.58	1.13	44.76	2.23E-11

8. The manuscript states: “In HUNT, UK Biobank, and FinnGen (Supplementary Table 12), we observed that the top lipid-lowering alleles at this locus were associated with reduced risk for CHD (Figure 2).” and “Notably, we observed consistently lower odds of T2D in both HUNT and FinnGen yielding, for rs5942648, a meta-analyzed OR = 0.97 (95% CI 0.96, 0.99; P = 1.4x10⁻⁵)”. These interpretations are overstatements, because for CHD, this is only true for the meta-analysis but not for the single studies, where the confidence interval includes the 1 for both UKBB and FinnGen. Similarly, for T2D, this is only true for the meta-analysis and the point estimates, but the confidence interval for HUNT again includes 1.

Author Response:

We have modified the Results to explicitly state that the effects are in the same direction instead of that there is evidence of association in all three studies. Given the sample sizes of HUNT and FinnGen, these studies have lower power to detect associations individually and we use meta-analysis to combine the evidence across the studies.

Manuscript Change:

“...we observed that the top lipid-lowering alleles at this locus showed a reduced risk for CHD (Figure 2).”

“Notably, we observed lower odds of T2D for rs5942648 (OR = 0.97; 95% CI 0.96, 0.99; P = 1.4x10⁻⁵) (Figure 2).”

Reviewer #5:

The authors present their data on an X-chromosome locus that associates with plasma lipid levels. Also provided are responses to a rigorous review of a previous version of the manuscript, for which the authors have tackled most issues. They have provided a very strong response and should be commended for doing so. On this basis many of the large issues have been dealt with but I have a few comments as detailed below.

Author Response:

We appreciate the enthusiastic response to our substantially revised manuscript.

1. A test for heterogeneity is not a substitution for an interaction test. In particular, heterogeneity estimated using Cochran’s Q between only 2 groups is fairly meaningless and yet still highly significant for triglycerides suggesting that a formal interaction test would confirm that the effects are not similar between men and women. Please formally test this interaction and alter the text accordingly if a significant interaction is found.

Author Response:

Thank you for this suggestion. We agree with the reviewer that the Cochran Q test of heterogeneity is a generally under powered approach to detect differences in effects. We have now performed an interaction test and observed that there is an interaction between rs5985504 and sex on triglycerides ($p=0.001$).

Manuscript Change:

Results:

“We observed similar associations for both males and females within TOPMed except male rs5985504-T carriers had greater decrease in triglycerides compared to female rs5985504-T carriers ($P_{\text{interaction}} = 0.001$) (**Supplementary Table 9**).”

Methods:

“we tested for the interaction between rs5985504-T and sex on log triglycerides adjusting for the same covariates as the main analysis.”

2. Colocalisation analyses could be extended to include BMI, T2D and CAD (rather than gene expression) to avoid over interpreting a single variant result from a PHEWAS that ignores LD. Given the established relationships at HMGCR and PCSK9 with lipids, CAD but not consistently with T2D and BMI this relationship could also be investigated for your variant through appropriate mediation analyses. This relationship with cardiometabolic traits is not clear enough to justify the title of the paper.

Author Response:

Our colocalization was performed to make insights into the causal gene in the locus. We correlated chrXq23 total cholesterol variants with MAF > 0.05 and $p < 0.05$ with BMI, T2D, and CAD in the UKBB. As seen corroborated with the results presented for the single variant, there was a correlation between the effect of the variants on total cholesterol and the effect of the variants on BMI, T2D, and CAD. We have been careful with the title of this paper – we note that the association is with lower atherogenic lipid concentrations ‘and’ favorable cardiometabolic indices. We specifically do not use ‘through,’ ‘mediates,’ etc.

The following text has been added to the results:

Manuscript Change:

“..., and a correlation between the effect sizes of variants on total cholesterol in the chrXq23 locus and the effect sizes of these variants on CAD ($r=0.25$), T2D ($r=0.33$), and BMI ($r=-0.34$; **Supplementary Figure 10**).

Supplementary Figure 10. Effects of ChrXq23 variants on total cholesterol (TC) vs effects of these variants on coronary artery disease (CAD), type 2 diabetes (T2D), and BMI.

Methods:

“To determine the correlation of the effect sizes of variants on total cholesterol in the chrXq23 locus and the effect sizes of these variants on CAD, T2D, and BMI, we performed analysis of chrXq23 variation on these three outcomes adjusting for age, age2, sex, genotyping array, and PCs in the UK Biobank, using the main effects model assuming X inactivation. We correlated effect sizes of total

cholesterol~variants with effect sizes of variants on CAD, T2D, or BMI limiting to total cholesterol variants with a MAF > 0.05 and a p-value < 0.05.”

3. Why, given there is no heterogeneity between ancestry in TOPMED did you only look at white-British in UK Biobank?

Author Response:

We have added the non-British white UK Biobank samples as a replication set. These results are now included in **Table 1**.

4. The methods for conditional analyses and how you estimate heterogeneity appear to be missing in the methods.

Author Response:

We have added the following text to the Methods:

Manuscript Change:

“Heterogeneity of effect sizes in the meta-analysis was determined through Cochran's Q and I^2 is reported.”

5. In conditional analyses you highlight your signal as only being partially explained by previously identified variants. i.e. still independent. However, given that the signal disappears for the published variants when you add your new variant I consider this new finding to be a refinement of the known loci rather than novel.

Author Response:

We have revised the first sentence of the Discussion to address that we report a refinement of a known loci.

Manuscript Change:

“Using deep-coverage next-generation sequencing of the X chromosome in the NHLBI TOPMed program, we identified a locus that was previously associated with lipids in primarily European ancestry datasets, which we now extend to diverse ancestries along with strong replication in two independent studies. In addition to replicating the associations of chrXq23 lipid-lowering alleles with reduced odds for CHD, we also observe associations with reduced T2D odds and favorable adiposity indices. These observations allow us to draw several conclusions about X chromosome genetic variation with blood lipid levels, as well as related cardiometabolic effects.”

Reviewer #4 (Remarks to the Author):

In their current revision, the authors have addressed all of my comments adequately, and the manuscript is much improved in my eyes with a clear description of methods and justified interpretation of results.

Reviewer #5 (Remarks to the Author):

A robust response from the authors that is pleasing to read, and I believe this has resulted in an improved manuscript. I have no further comments.

RESPONSE TO REVIEWERS' COMMENTS

Reviewer #4 (Remarks to the Author):

In their current revision, the authors have addressed all of my comments adequately, and the manuscript is much improved in my eyes with a clear description of methods and justified interpretation of results.

Author Response:

We appreciated the constructive feedback to improve the manuscript.

Reviewer #5 (Remarks to the Author):

A robust response from the authors that is pleasing to read, and I believe this has resulted in an improved manuscript. I have no further comments.

Author Response:

We thank the Reviewer for their input, which has improved the manuscript.